# CursorCore: Assist Programming through Aligning Anything

**Hao Jiang** [1]  **Qi Liu** [1 2]  **Rui Li** [1]  **Shengyu Ye** [1]  **Shijin Wang** [3]

## Abstract

Large language models have been successfully applied to programming assistance tasks, such as code completion, code insertion, and instructional code editing. However, these applications remain insufficiently automated and struggle to effectively integrate various types of information during the programming process, including coding history, code context, and user instructions. In this work, we propose a new framework that comprehensively integrates these information sources, and collect data to train models and evaluate their performance. Firstly, to thoroughly evaluate how well models align with different types of information and the quality of their outputs, we introduce a new benchmark, APEval (Assist Programming Eval), to comprehensively assess the performance of models in programming assistance tasks. Then, for data collection, we develop a data generation pipeline, Programming-Instruct, which synthesizes training data from diverse sources, such as GitHub and online judge platforms. This pipeline can automatically generate various types of messages throughout the programming process. Finally, using this pipeline, we generate 219K samples, fine-tune multiple models, and develop the CursorCore series. We show that CursorCore outperforms other models of comparable size. This framework unifies applications such as inline chat and automated editing, contributes to the advancement of coding assistants.

## 1. Introduction

Since the rise of large language models (LLMs), AI-assisted programming technology has developed rapidly, with many powerful LLMs being applied in this field (Zan et al., 2022;

Liang et al., 2024; Yang et al., 2024). The technology mainly takes two forms. One form involves completing a specified code snippet at the end or inserting corresponding code at a designated position, typically accomplished by foundation models (Chen et al., 2021; Bavarian et al., 2022) that support relevant input formats. The other form involves generating or editing code snippets based on natural language instructions or reflections through interaction with the environment, usually carried out by instruction models that have been further aligned (Shinn et al., 2023; Cassano et al., 2023b; Muennighoff et al., 2024; Paul-Gauthier, 2024). Figure 1 shows simple examples of these forms.

However, in practical applications, neither the completion or insertion mode nor the instruction-based mode is perfect. The completion or insertion mode generates based on the current code context, but in actual coding, we are continuously editing the code rather than just completing and inserting. We prefer that the model predicts the upcoming edits, as neither completion nor insertion accurately reflects the coding process, and requires programmers to perform additional operations. The instruction-based mode allows for code editing, but it also has drawbacks, such as writing prompts for specific tasks may be slower or challenging. The process is not automated enough, programmers would prefer a model that can proactively predict future changes without needing extra prompts. In our view, the core issue lies in the limitations of the input and output in both forms of programming assistance. These forms either just align the output with the current code context, limiting completion or insertion instead of editing, or align the output with the user's natural language instructions. However, to effectively assist with programming, an AI programming assistant needs to utilize anything throughout the programming process. It should be capable of aligning with the history of code changes, the current content of the code, and any instructions provided by the user, predicting the required responses and corresponding changes, reducing any actions required by users.

To solve these issues, in this paper, we introduce a new framework of AI-assisted programming task: Assistant-Conversation to align anything during programming process. To comprehensively evaluate the alignment of models with different information in the programming process and the quality of the corresponding outputs, we propose a new

---

[1]State Key Laboratory of Cognitive Intelligence, University of Science and Technology of China [2]Institute of Artificial Intelligence, Hefei Comprehensive National Science Center [3]iFLYTEK Co., Ltd. Correspondence to: Qi Liu <qiliuql@ustc.edu.cn>.

*Proceedings of the 42$^{nd}$ International Conference on Machine Learning*, Vancouver, Canada. PMLR 267, 2025. Copyright 2025 by the author(s).

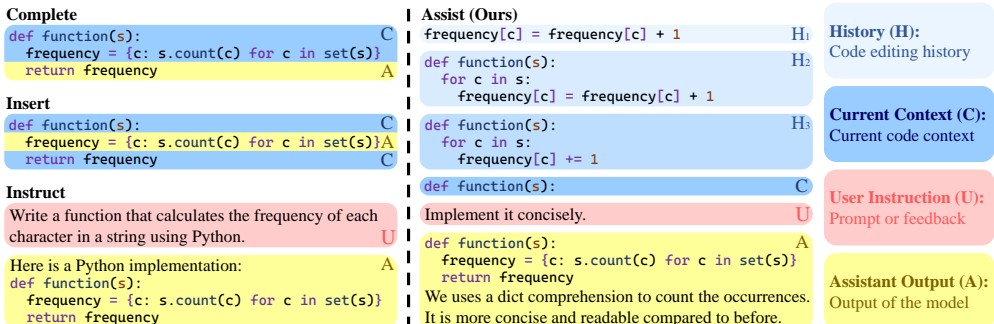

*Figure 1.* Different forms of programming assistance. The common uses of current LLMs are shown on the left. Our framework is shown on the right.

benchmark, APEval (Assist Programming Eval), to comprehensively assess the performance of models in assisting programming. For the Assistant-Conversation framework, we build a data generation pipeline, Programming-Instruct, to synthesize corresponding training data from various data sources. This data generation method can produce any types of messages throughout the programming process, without any additional human annotation and does not rely on specific models. We use it to generate 219K data points and use them to fine-tune multiple models, resulting in the Cursor-Core series. These models achieve state-of-the-art results when compared with other models of comparable size.

In conclusion, our main contributions are:

- Assistant-Conversation: A new framework to align anything during programming process.

- Programming-Instruct: Data synthesis pipeline to produce any types of messages throughout the programming process, and 219K data collected using it.

- APEval: A comprehensive benchmark for assessing the ability to utilize various types of information to assist programming.

- CursorCore: One of the best model series with the same number of parameters for AI-assisted programming tasks.

Code, models and data are freely available at https://github.com/TechxGenus/CursorCore.

## 2. Assistant-Conversation: New Conversation Framework for Programming Assistants

In this section, we introduce a new conversational framework, Assistant-Conversation, aimed at simplifying the programming process[1]. The framework leverages all available

___
[1] In this work, "conversation" refers to the common format used in LLM generation, rather than multi-turn dialogues.

information during programming to streamline work for programmers. By precisely defining various types of information and their formats, Assistant-Conversation directly aligns with the input and output requirements of applications such as automated editing and inline chat. This framework facilitates model alignment, enabling fast and accurate generation and parsing.

### 2.1. Framework Formulation

We introduce the elements of Assistant-Conversation: System ($S$), History ($H$), Current Context ($C$), User Instruction ($U$), and Assistant Output ($A$). $A$ represents the output of the model, while the inputs consist of $S$, $H$, $C$, $U$. Figures 1 and 2 shows several examples of them. These definitions will be referenced throughout the rest of this work.

**System $S$ (Optional)** The system instruction provided to the model at the beginning, which configures the answering style, overall task description and other behaviors. In this work, we fix it to a simple "You are a helpful programming assistant." and omit it from the subsequent discussion.

**History $H$ (Optional)** The program's editing history, consisting of multiple pieces of code. These may include several snippets or may not be present at all. We refer to them as $H_1, \cdots, H_n$.

**Current Context $C$** The code context currently being processed, along with temporary information like cursor position or selected code area.

**User Instruction $U$ (Optional)** User instructions related to the code, either written by the programmer or generated as feedback based on interactions with external environments (such as a code interpreter).

**Assistant Output $A$** The output of the model, consists of modified code and chat-style interaction with the program-

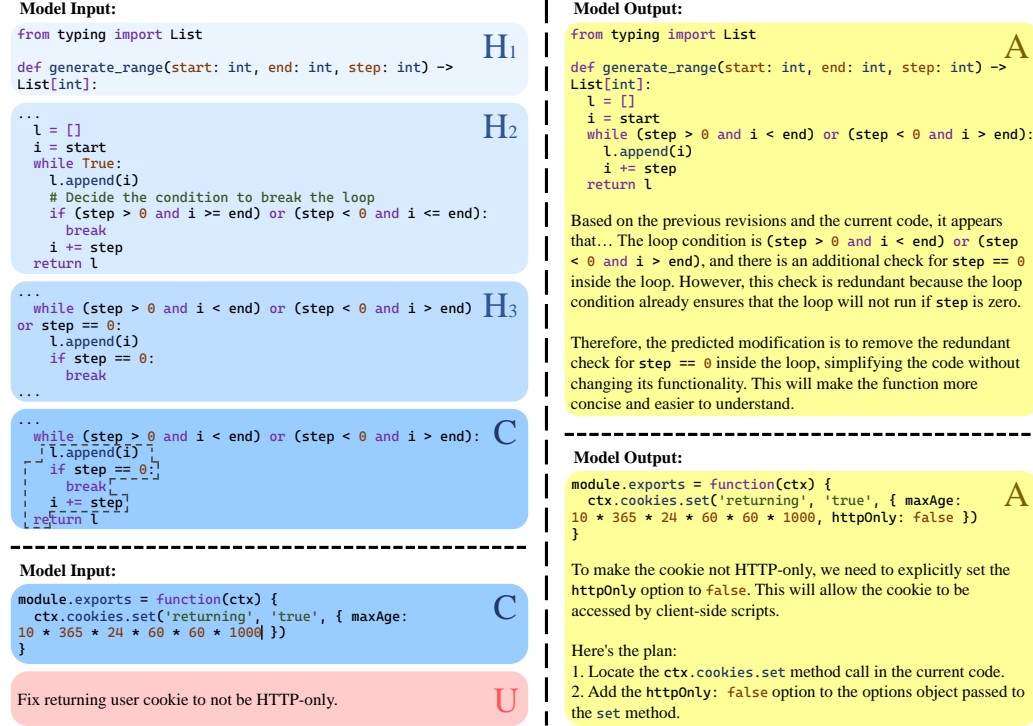

*Figure 2.* Examples of Assistant-Conversation from our training data. The top example demonstrates predicting the corresponding edits and explanations based on historical edits and the current code context. The bottom example demonstrates predictions based on the current code and user instructions.

mer. In this work, we mainly focus on the prediction of modified code.

## 2.2. Comparisons of Assistant-Conversation

**Completion and insertion modes face challenges when modeling both *C* and *H*** Although they can utilize *C*, they fail to capture *H*, limiting the modeling of future changes in *C*, and are incapable of deleting or editing code. Although user instructions and reflection information can be used through comments and assert statements, this capability is weak and unstable.

**Chat models are not ideal for all programming assistance tasks** These models focus on user input rather than the code content, while the input should primarily be centered on *C* instead of just user instructions. In traditional conversational frameworks, the sole input source is *U*, which works for chatbots but not for application assistants. Input sources should include *C*, *H*, and *U*, as both *H* and *U* are related to *C*. Although instruction models can represent the interaction history between users and assistants, they struggle to capture the historical changes in the application's content. Prompt engineering can integrate some of this information into existing models, but the impact is limited. Constructing prompts with numerous tokens increases cost and reduces

efficiency, and models may also lack alignment and proper training for such inputs.

**Our framework addresses these issues** We use multiple input sources to harness all relevant information from the programming process. For the output, we divide it into two parts: modified code and chat-style communication with the programmer, aligning with the common practices of users. When the user only requires responses based on *U*, similar to instruction models, we can omit *H* and *C*, suppress code modifications, and provide only chat output to ensure compatibility with past chat modes.

## 2.3. Specifications and Implementation

To represent a piece of code like *C*, we can either use it directly or wrap it in a markdown code block. However, representing code changes, such as *H* or changes in *A*, is more complex. We can either use the whole code, patches that alter the code, or records of both the modification locations and the specific changes. Some methods work well but experience issues when handling longer texts, such as outputting the entire modified code, which can be slow. Other methods output minimal content, like providing only the modification locations and changes. These are faster but still not optimal in terms of performance. We represent

code changes in the experiments of the main body using the whole code format, and we investigate different ways to represent these modifications, as detailed in Appendix B. Additionally, we explore methods for compressing historical code changes in Appendix I.

In some cases, programmers assign assistants to focus on specific areas of code. They might use the cursor to mark a general location or directly select a range of code, as shown in Figure 2. We handle this by treating them as special tokens (see Appendix F for further details).

We structure conversations in the order of *S-H-C-U-A* to match the actual workflow. This mirrors the chronological sequence in which information is generated during the programming process. By doing so, we maximize prefix overlap across multiple requests, utilizing prefix caching to reduce redundant kv-cache computations and improve efficiency (Zheng et al., 2023a). *A* is organized in code-chat order, prioritizing code edits due to their importance in real-time applications where speed is crucial.

## 3. APEval: Benchmark for Assisted Programming

### 3.1. Benchmark overview

Past benchmarks assessing LLM code capabilities have effectively evaluated tasks like program synthesis (Chen et al., 2021; Austin et al., 2021), code repair (Muennighoff et al., 2024; Jimenez et al., 2024), and instructional code editing (Cassano et al., 2023b; Paul-Gauthier, 2024; Guo et al., 2024b). However, they fall short in fully assessing how models use various types of information to assist in programming. This gap calls for a new benchmark.

*Table 1.* APEval Statistics and breakdown of tasks by information type.

| APEval Statistics | | |
|---|---|---|
| Python | 164 Samples | |
| Multilingual | 984 Samples | |
| Language | Python, C++, Java, JavaScript, Go, Rust | |
| **Details** | **Mean** | **Max** |
| Snippets (H) | 2.8 | 10 |
| Lines (H\|C\|U) | 21.7 \| 8.4 \| 3.2 | 139 \| 31 \| 19 |
| Chars (H\|C\|U) | 0.6K \| 0.3K \| 0.2K | 5.1K \| 1.4K \| 1.2K |

As discussed in Section 2.1, programming assistance can involve different types of information, with *H* and *U* being optional. Thus, there are four possible combinations of information: *H, C, U*; *H, C*; *C, U*; and only *C*. HumanEval (Chen et al., 2021) is a well-known benchmark for evaluating code completion. It has been extended to assess other tasks such as code insertion (Bavarian et al., 2022), instruction-based

tasks (CodeParrot, 2023; Muennighoff et al., 2024), and multilingual generation (Zheng et al., 2023b; Cassano et al., 2023a). We refer to these works and further extend it to comprehensively evaluate the model's ability to assist programming. We randomly categorize each task into one of the four types, then manually implement the functions and simulate the potential instructions that programmers might give to an LLM during the process, collecting all interactions. We invite programmers with varying levels of experience to annotate the data. After processing, we get the new benchmark, Assist Programming Eval (APEval), which contains approximately 1K multilingual samples. Detailed statistics are shown in Table 1. Specific details regarding the collection process and examples of our benchmark can be found in Appendix C, which includes detailed human annotation rubric and results.

### 3.2. Evaluation Process and Metrics

In all tasks, we use the classic Pass@1 metric to execute the generated code, which is the simplest version of the Pass@k metric (Chen et al., 2021). Since APEval is an extension of HumanEval, we evaluate its Python version using the test set created by EvalPlus (Liu et al., 2023) and assess its other language versions using bigcode-evaluation-harness (Ben Allal et al., 2022). We set the Python version as the default version for evaluation, and report the results from both the basic and extra tests. We provide the model with relevant information during the programming process, and the model immediately returns the modified code. Some methods may improve performance by increasing the number of output tokens to model the thinking process; we discuss this further in Appendix G.

## 4. Programming-Instruct: Collect any data during programming

To align models with programming-related data, relevant training data must be collected. While large amounts of unsupervised code (Kocetkov et al., 2023) and instruction data (Wei et al., 2023b; Luo et al., 2024b) have been gathered, there remains a significant lack of data on the coding process. Manually annotating the coding process is expensive, so we propose Programming-Instruct, a method to automate this data collection.

### 4.1. Data Sources

To ensure both quality and diversity in the coding process data, we collect information from three different sources: AI Programmer, Git Commit, and Online Judge submission.

**AI Programmer**  For each code snippet, we use LLMs to generate the corresponding coding history. Since human

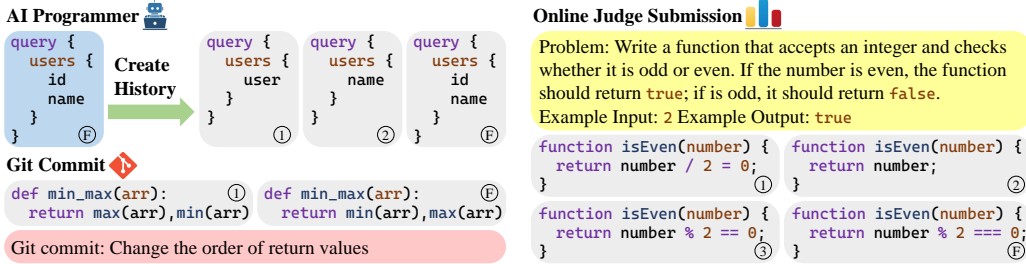

*Figure 3.* Samples from AI Programmer, Git Commit and Online Judge Submission.

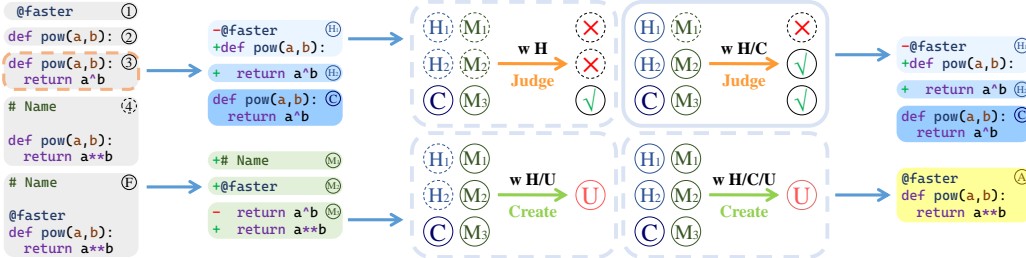

*Figure 4.* Data processing pipeline. The randomly selected time point is the third, data type is *H* and *C*.

coding approaches vary widely, we utilize several LLMs, each guided by three distinct prompts, representing novice, intermediate, and expert programmers. The LLMs then return their version of the coding process. Prompts used are shown in Appendix O.

**Git Commit** Some software can automatically track changes, such as Git. We use Git Commit data from Github, which captures users' code edits and modification histories.

**Online Judge Submission** Many online coding platforms like Leetcode and Codeforces allow users to submit code for execution and receive feedback. During this process, users continuously modify their code until it is finalized. We also make use of this data.

Through these sources, we obtain a large number of samples, each consisting of multiple code snippets. The last snippet in each sample is referred to as the final snippet (*F*). Examples of data sources are shown in Figure 3.

### 4.2. Data Processing

After collecting programming processes, we process them to meet the requirements of Assistant-Conversation. Figure 4 shows the steps of data processing. First, we randomly select a time point in the coding process, referred to as *C*. As mentioned in Section 2.1, *H* and *U* are optional, we need to collect four types of data distinguished according to input data types: *H*, *C*, *U*; *H*, *C*; *C*, *U*; and only *C*. For each sample, we randomly designate one type. If the selected type includes *H*, We use the preceding edits of *C* as the

historical records *H*.

We then handle each type of data based on whether *U* is available. For cases without *U*, we segment the changes from *C* to *F* based on continuity, referring to them as *M*, and let LLMs analyze and then judge whether each segment of *M* aligns with user's purpose through principle-driven approaches (Bai et al., 2022; Sun et al., 2023; Lin et al., 2024). This approach accounts for ambiguity in user intent when inferring from *H* or *C*. For example, if a programmer actively adds some private information at the beginning of the code without it being mentioned in the previous records, LLMs should not predict this change. We discard segments deemed irrelevant, and merge the remaining ones as outputs that models need to learn to predict. For cases with *U*, we follow the instruction generation series methods (Wang et al., 2023b; Wei et al., 2023b; Luo et al., 2024b) by inputting both the historical edits and current code into the LLM, prompting it to generate corresponding instructions.

In addition to the above, we model selected code regions, cursor positions, and make LLMs create chat-style interactions with users. Further details are provided in Appendix D.

## 5. CursorCore: Fine-tune LLMs to align anything

### 5.1. Base models

We fine-tune existing base LLMs to assist with programming tasks. Over the past few years, many open-source foundation models have been trained on large code corpora

*Table 2.* Statistics of our training data.

|  | Sample Num | Language Num | History Snippets Mean / Max | Input Length Mean / Max | Output Length Mean / Max |
|---|---|---|---|---|---|
| AI Programmer | 70.9K | - | 2.0 / 17 | 0.6K / 25K | 1.0K / 5.2K |
| Git Commit | 88.0K | 14 | 1.5 / 15 | 1.5K / 19.9K | 1.4K / 5.2K |
| Online Judge Submission | 60.5K | 44 | 3.8 / 96 | 4.8K / 357.2K | 1.9K / 35.1K |

sourced from GitHub and other platforms, demonstrating strong performance in coding. We choose the base versions of Deepseek-Coder (Guo et al., 2024a), Yi-Coder (AI et al., 2024) and Qwen2.5-Coder (Hui et al., 2024) series, as fine-tuning is generally more effective when applied to base models rather than instruction models. After training, we refer to them as CursorCore-DS, CursorCore-Yi and CursorCore-QW2.5 series. Deepseek-Coder has achieved state-of-the-art performance on numerous coding-related benchmarks over the past year, gaining wide recognition. Yi-Coder and Qwen2.5-Coder are the most recently released models at the start of our experiments and show the best performance on many benchmarks for code now. These models are widely supported by the community, offering a good balance between size and performance, making them suitable for efficient experimentation. For ablation experiments, we use the smallest version, Deepseek-Coder-1.3B, to accelerate the process. We use a chat template adapted from ChatML (OpenAI, 2023) to model Assistant-Conversation during training, as detailed in Appendix M. Training details can be found in Appendix E.

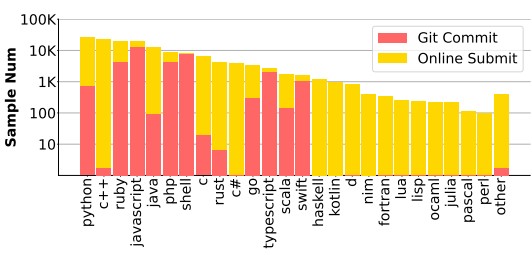

*Figure 5.* Distribution of programming language in the training data.

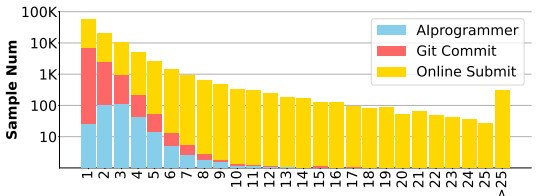

*Figure 6.* Distribution of history snippet counts in the training data.

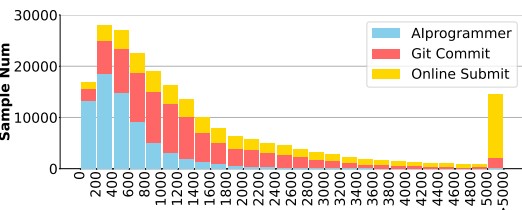

*Figure 7.* Distribution of input lengths in the training data.

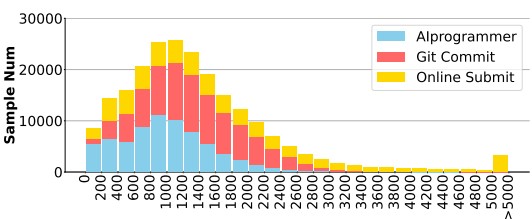

*Figure 8.* Distribution of output lengths in the training data.

### 5.2. Training data

We use Programming-Instruct to collect data. For AI Programmer, we gather code snippets from datasets such as the Stack (Kocetkov et al., 2023) and OSS-Instruct (Wei et al., 2023b), then prompt LLMs to generate the programming process. For Git Commit data, we collect relevant information from EditPackFT (Cassano et al., 2023b) (a filtered version of CommitPackFT (Muennighoff et al., 2024)) and further refine it through post-processing and filtering. Regarding Online Judge Submission data, we source the programming process from the Codenet dataset (Puri et al., 2021). First, we group all submissions by user for each problem, then exclude invalid groups without correct submissions to obtain complete programming processes. These are then fed into the processing pipeline to generate the final training data. In total, we accumulate 219K samples, with detailed statistics and distributions shown in Tables 2 and 3 and Figures 5 to 8. AI Programmer data has the shortest average length, while Online Judge Submission data has the longest. To ensure compatibility with previous chatbot-style interactions and further improve model performance, we also incorporate the Evol-Instruct dataset (ISE-UIUC, 2023) collected using the GPT series (Ouyang et al., 2022), which has been widely recognized for its high quality during train-

ing. Following StarCoder's data processing approach (Li et al., 2023), we decontaminate our training data.

During data collection, we randomly utilize two powerful open-source LLMs: Mistral-Large-Instruct (Mistral-AI, 2024b) and Deepseek-Coder-V2-Instruct (DeepSeek-AI et al., 2024). These models have demonstrated performance comparable to strong closed-source models like GPT-4o across many tasks, and are currently the only two open-source models scoring over 90% on the classic HumanEval benchmark at the start of our experiment. Additionally, they are more cost-effective and offer easier reproducibility than GPT-4o. For Mistral-Large-Instruct, we quantize the model using the GPTQ (Frantar et al., 2022) algorithm and deploy it locally with SGLang (Zheng et al., 2023a) and Marlin kernel (Frantar et al., 2024) on 4 Nvidia RTX 4090 GPUs. For Deepseek-Coder-V2-Instruct, we use its official API for integration.

*Table 3.* The proportion of four combinations of information during programming in our training data.

|                         | C    | H, C | C, U | H, C, U |
|-------------------------|------|------|------|---------|
| AI Programmer           | 24.1 | 22.2 | 25.4 | 28.3    |
| Git Commit              | 25.9 | 20.0 | 28.0 | 26.1    |
| Online Judge Submission | 27.5 | 19.7 | 29.4 | 23.4    |

## 6. Evaluation and Results

In this section, we evaluate the CursorCore models. We begin by describing the experimental setup and then present and analyze the results.

### 6.1. Experimental setup

We conduct the data selection ablation and primary evaluation on our APEval benchmark, and provide results on well-known benchmarks such as Python program synthesis, automated program repair, and instructional code editing, which are detailed in Appendix J. We choose prominent open-source and closed-source LLMs as our baselines. For all benchmarks, we use greedy decoding to generate evaluation results. CursorCore natively supports various inputs in APEval, whereas base and instruction LLMs require additional prompts for effective evaluation. We design few-shot prompts separately for base and instruction models, as detailed in Appendix N. Data selection ablation can be found in Appendix H.

### 6.2. Evaluation results on APEval

In Table 4, we present the results of evaluating CursorCore series models and other LLMs on the Python version of APEval. The results for multilingual versions can be found

in Appendix L. It includes both the average results and the results across four different types of information within the benchmark, each item in the table is the score resulting from running the base tests and extra tests. We also report the evaluation results of other well-known models, which can be found in Appendix K.

**CursorCore outperforms other models of comparable size** CursorCore consistently outperforms other models in both the 1B+ and 6B+ parameter sizes. It achieves the highest average score, with the best 1B+ model surpassing the top scores of other models by 10.4%, and even by 11.5% when running extra tests. Similarly, the best 6B+ model exceeds by 4.3%, and by 3.0% in the case of extra tests. Additionally, across various information types, CursorCore consistently demonstrates optimal performance among all similarly sized models.

**Instruction models mostly outperform base models** For most model series, instruction-tuned models outperform their corresponding base models, as instruction fine-tuning generally enhances model capabilities (Ouyang et al., 2022; Longpre et al., 2023). The only exception observed in our experiments is the latest model, Qwen2.5-Coder. Its base model achieves a very high score, while the instruction-tuned model performes worse. We attribute the base model's high performance to its extensive pre-training, which involved significantly more tokens than previous models (Hui et al., 2024). This training on a wide range of high-quality data grants it strong generalization abilities, enabling it to effectively handle the newly defined APEval task format. In contrast, the instruction-tuned model is not specifically aligned with this task, leading to a decrease in its APEval score. This highlights the challenges of aligning models with numerous diverse tasks, especially small models.

**Performance difference between general and code LLMs is strongly related to model size** In 1B+ parameter models, general LLMs significantly underperform code LLMs. Even the best-performing general model scores over 10% lower compared to the best-performing code model, despite having more parameters. For models with 6B+ parameters, while general LLMs still lag behind code LLMs, the performance gap narrows considerably, with general LLMs even surpassing in certain cases involving specific information types. When it comes to 10B+ models, the performance difference between general and code LLMs becomes negligible. We think that smaller models, due to their limited parameter capacity, tend to focus on a single domain, such as programming assistance, while larger models can encompass multiple domains without compromising generalizability.

**Gap between closed models and the best open models is smaller** Historically, open-source models significantly

Table 4. Evaluation results of LLMs on APEval.

| Model | C | H, C | C, U | H, C, U | Avg. |
|---|---|---|---|---|---|
| **Closed Models** | | | | | |
| GPT-4o | 68.3 (63.4) | 61.0 (56.1) | 75.6 (75.6) | 56.1 (53.7) | 65.2 (62.2) |
| **10B+ Models** | | | | | |
| Codestral-V0.1-22B | 68.3 (56.1) | 41.5 (41.5) | 75.6 (73.2) | 48.8 (46.3) | 58.5 (54.3) |
| DS-Coder-33B-Inst | 63.4 (56.1) | 56.1 (48.8) | 70.7 (63.4) | 51.2 (48.8) | 60.4 (54.3) |
| Qwen2.5-72B-Inst | 73.2 (68.3) | 53.7 (51.2) | 78.0 (70.7) | 56.1 (56.1) | 65.2 (61.6) |
| Mistral-Large-123B-Inst | 65.9 (58.5) | 56.1 (46.3) | 73.2 (68.3) | 48.8 (48.8) | 61.0 (55.5) |
| DS-Coder-V2-236B-Inst | 78.0 (65.9) | 48.8 (43.9) | 68.3 (61.0) | 53.7 (48.8) | 62.2 (54.9) |
| **6B+ Models** | | | | | |
| Llama-3.1-8B-Inst | 24.4 (24.4) | 31.7 (29.3) | 53.7 (51.2) | 39.0 (34.1) | 37.2 (34.8) |
| Gemma-2-9B-It | 56.1 (53.7) | 41.5 (36.6) | 51.2 (46.3) | 36.6 (29.3) | 46.3 (41.5) |
| DS-Coder-6.7B-Base | 29.3 (24.4) | 26.8 (22.0) | 41.5 (31.7) | 22.0 (19.5) | 29.9 (24.4) |
| DS-Coder-6.7B-Inst | 56.1 (53.7) | 41.5 (36.6) | 70.7 (61.0) | 34.1 (29.3) | 50.6 (45.1) |
| Yi-Coder-9B | 29.3 (26.8) | 26.8 (22.0) | 17.1 (17.1) | 29.3 (26.8) | 25.6 (23.2) |
| Yi-Coder-9B-Chat | 56.1 (51.2) | 39.0 (36.6) | 73.2 (70.7) | 36.6 (36.6) | 51.2 (48.8) |
| Qwen2.5-Coder-7B | 56.1 (53.7) | 41.5 (36.6) | 65.9 (56.1) | 31.7 (29.3) | 48.8 (43.9) |
| Qwen2.5-Coder-7B-Inst | 22.0 (19.5) | 46.3 (39.0) | 75.6 (65.9) | 41.5 (39.0) | 46.3 (40.9) |
| CursorCore-DS-6.7B | **68.3 (63.4)** | 41.5 (39.0) | 68.3 (63.4) | 36.6 (31.7) | 53.7 (49.4) |
| CursorCore-Yi-9B | 53.7 (53.7) | **46.3 (43.9)** | 75.6 (**68.3**) | 43.9 (36.6) | 54.9 (50.6) |
| CursorCore-QW2.5-7B | 65.9 (61.0) | 41.5 (39.0) | 65.9 (63.4) | **48.8 (43.9)** | **55.5 (51.8)** |
| **1B+ Models** | | | | | |
| Llama-3.2-3B-Instruct | 14.6 (14.6) | 22.0 (19.5) | 29.3 (26.8) | 34.1 (31.7) | 25.0 (23.2) |
| Gemma-2-2B-It | 14.6 (14.6) | 22.0 (19.5) | 29.3 (26.8) | 34.1 (31.7) | 25.0 (23.2) |
| DS-Coder-1.3B-Base | 0.0 (0.0) | 12.2 (12.2) | 17.1 (12.2) | 19.5 (14.6) | 12.2 (9.8) |
| DS-Coder-1.3B-Inst | 39.9 (36.6) | 39.0 (36.6) | 39.0 (29.3) | 34.1 (34.1) | 37.8 (34.1) |
| Yi-Coder-1.5B | 2.4 (0.0) | 2.4 (2.4) | 14.6 (14.6) | 12.2 (7.3) | 7.9 (6.1) |
| Yi-Coder-1.5B-Chat | 31.7 (31.7) | 4.9 (4.9) | 51.2 (41.5) | 26.8 (22.0) | 28.7 (25.0) |
| Qwen2.5-Coder-1.5B | 43.9 (36.6) | 26.8 (26.8) | 51.2 (41.5) | 36.6 (34.1) | 39.6 (34.8) |
| Qwen2.5-Coder-1.5B-Inst | 14.6 (14.6) | 17.1 (14.6) | 43.9 (34.1) | 31.7 (29.3) | 26.8 (23.2) |
| CursorCore-DS-1.3B | 36.6 (31.7) | 39.0 (31.7) | 53.7 (46.3) | 26.8 (22.0) | 39.0 (32.9) |
| CursorCore-Yi-1.5B | **46.3** (39.0) | 34.1 (29.3) | **68.3** (58.5) | 36.6 (34.1) | 46.3 (40.2) |
| CursorCore-QW2.5-1.5B | **46.3 (43.9)** | **48.8 (43.9)** | 65.9 (**61.0**) | **39.0 (36.6)** | **50.0 (46.3)** |

lag behind closed-source models, like those in the GPT series, leading to a preference for closed-source models in synthetic data generation and other applications (Taori et al., 2023; Xu et al., 2023). However, with the continuous advancement of open-source LLMs, increasingly powerful models have emerged. On APEval, the best open-source models—such as Qwen2.5-72B-Instruct, Mistral-Large-Instruct, and Deepseek-Coder-V2-Instruct—demonstrate performance that closely approaches that of the leading GPT series model, GPT-4o. This indicates that the performance gap between open-source and closed-source LLMs has considerably narrowed, encouraging the development of more interesting applications based on open-source LLMs. Despite this progress, GPT-4o remains more comprehensive than open-source LLMs. It utilizes $H$ far more effectively than any other model, demonstrating its strong capability to

process and align with various types of information. This is an area where open-source LLMs still need to improve.

# 7. Conclusion

This work explores how LLMs can maximize the use of any available information during programming process to assist coding. We introduce Assistant-Conversation to model the diverse types of information involved in programming. We present APEval, a new benchmark that includes various historical edits and instructions, providing a comprehensive evaluation of the model's programming assistance capabilities. Additionally, we propose Programming-Instruct, which is designed to collect data for training LLMs to assist programming, along with their corresponding data sources. Furthermore, we train CursorCore, which demonstrate out-

standing performance in assisting programming tasks while achieving a good balance between efficiency and cost. We also conduct extensive ablation experiments and analyzes. Beyond enhancing traditional approaches of programming assistance, we plan to extend this approach to support models capable of assisting with repository-level development as well as other applications.

## Acknowledgments

This research was partially supported by grants from the Joint Research Project of the Science and Technology Innovation Community in Yangtze River Delta (No. 2023CSJZN0200), the National Natural Science Foundation of China (62337001), the Key Technologies R & D Program of Anhui Province (No. 202423k09020039) and the Fundamental Research Funds for the Central Universities.

## Impact Statement

This paper presents work whose goal is to advance the field of Machine Learning. There are many potential societal consequences of our work, none which we feel must be specifically highlighted here.

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

# A. Related Work

## A.1. AI-Assisted Programming

AI-assisted programming has a long history, encompassing various tasks such as clone detection (Lu et al., 2021), knowledge tracing (Liu et al., 2019; Li et al., 2022; Gao et al., 2025), data mining (Wang et al., 2023a; Zhang et al., 2024c), code retrieval (Li et al., 2024b;c; 2025), code summarization (Jiang et al., 2025; Sun et al., 2024), program synthesis (Chen et al., 2021; Austin et al., 2021), automatic program repair (Gulwani et al., 2016; Zhao et al., 2024), code editing (Wei et al., 2023a), and code optimization (Shypula et al., 2024). These tasks attempt to incorporate a wide range of information into their processes, such as historical edits (Gupta et al., 2023; Zhang et al., 2022) and user instructions (Cassano et al., 2023b). In the past, however, they were typically addressed by custom-built models, which were difficult to scale across different tasks and types of information. With the rise of LLMs, AI-assisted programming increasingly leverages LLMs to handle multiple types of tasks simultaneously. Numerous high-quality open-source and closed-source products, such as Continue (Continue-Dev, 2024), Aider (Paul-Gauthier, 2024), Copilot (Github-Copilot, 2022) and Cursor (Cursor-AI, 2023), are based on this approach.

## A.2. Code Models

Recently, LLMs have attracted significant attention in the research community for their impact on enhancing various aspects of code intelligence. Open-source code LLMs like CodeLlama (Rozière et al., 2023; Touvron et al., 2023), Deepseek-Coder (Guo et al., 2024a; DeepSeek-AI et al., 2024), StarCoder (Li et al., 2023; Lozhkov et al., 2024), Codegemma (Team et al., 2024), Codestral (Mistral-AI, 2024a), Codegeex (Zheng et al., 2023b), Yi-Coder (AI et al., 2024), and Qwen-Coder (Hui et al., 2024) have made substantial contributions by utilizing large code corpora during training. Some models, such as WizardCoder (Luo et al., 2024b), OctoCoder (Muennighoff et al., 2024), CodeLlama-Instruct, Deepseek-Coder-Instruct, MagiCoder (Wei et al., 2023b), Yi-Coder-Chat, and Qwen-Coder-Instruct, have been fine-tuned using instruction data collected through methods like Self-Instruct (Wang et al., 2023b; Taori et al., 2023), Evol-Instruct, and OSS-Instruct. These models are specifically trained on code-related instructions, improving their ability to follow coding instructions. They have made significant breakthroughs in tasks like code completion and editing.

## A.3. Code Benchmarks

HumanEval (Chen et al., 2021) is one of the most well-known benchmarks in the code domain, featuring several variants that extend it to different programming languages, extra tests, and broader application scenarios. Other notable benchmarks include MBPP (Austin et al., 2021) for program synthesis, DS1000 (Lai et al., 2022) for data science tasks, SWE-Bench (Jimenez et al., 2024) for real-world software engineering problems, and CanItEdit / CodeEditorBench (Cassano et al., 2023b; Guo et al., 2024b) for code editing. Additionally, LiveCodeBench (Jain et al., 2024) focuses on contamination-free evaluations, while ClassEval(Du et al., 2023), Bigcodebench (Zhuo et al., 2024) and Naturecodebench (Zhang et al., 2024b) provide comprehensive program synthesis assessments. CRUXEval (Gu et al., 2024) targets reasoning, CrossCodeEval (Ding et al., 2023) focuses on repository-level code completion, and Needle in the code (Hui et al., 2024) is designed for long-context evaluations.

# B. Code modification representation

As discussed in Section 2.3, there are various ways to represent code modifications. Many previous works have explored techniques for instruction-based code editing (Wei et al., 2023a; Muennighoff et al., 2024; Paul-Gauthier, 2024; Sweep-AI, 2024). We build upon these works with the following formats, as shown in Figure 9:

**Whole file format (WF)**   We use the entire code, allows for a straightforward representation of the modifications. However, when only small parts of the code are changed, this method leads to redundancy, especially for long code files. Certain mitigation can be achieved through technologies such as retrieval-based speculative decoding (Yang et al., 2023; He et al., 2024).

**Unified diff format (UD)**   The diff format is a common way to represent code changes, widely adopted for its efficiency and readability. Among various diff formats, unified diff is one of the most popular, as it efficiently shows code changes while reducing redundancy. It is commonly used in software tools such as git and patch.

**Location-and-change format (LC)**   To further reduce redundancy, we consider further simplify the diff formats by showing only the location and content of the changes. The location is based on line numbers. Some reports indicate that LLMs often struggle with localization, so we insert line numbers into the code to assist them.

**Search-and-replace format (SR)**   Another option is to eliminate the need for localization altogether by simply displaying the part to be modified alongside the updated version. This format eliminates the need for line numbers.

We conduct experiments using Deepseek-Coder-1.3B with these formats. For quick experiments, we train the model on data generated by AI Programmer. We then evaluate their performance on APEval, with results shown in Figure 10. In programming assistance tasks, where real-time performance is critical, such as in tasks like auto completion or editing, the generation speed becomes particularly important. The number of tokens in both input and output directly affects the model's speed, and the editing format greatly impacts the token count. Therefore, we also report the average input-output token count for each format in Figure 11.

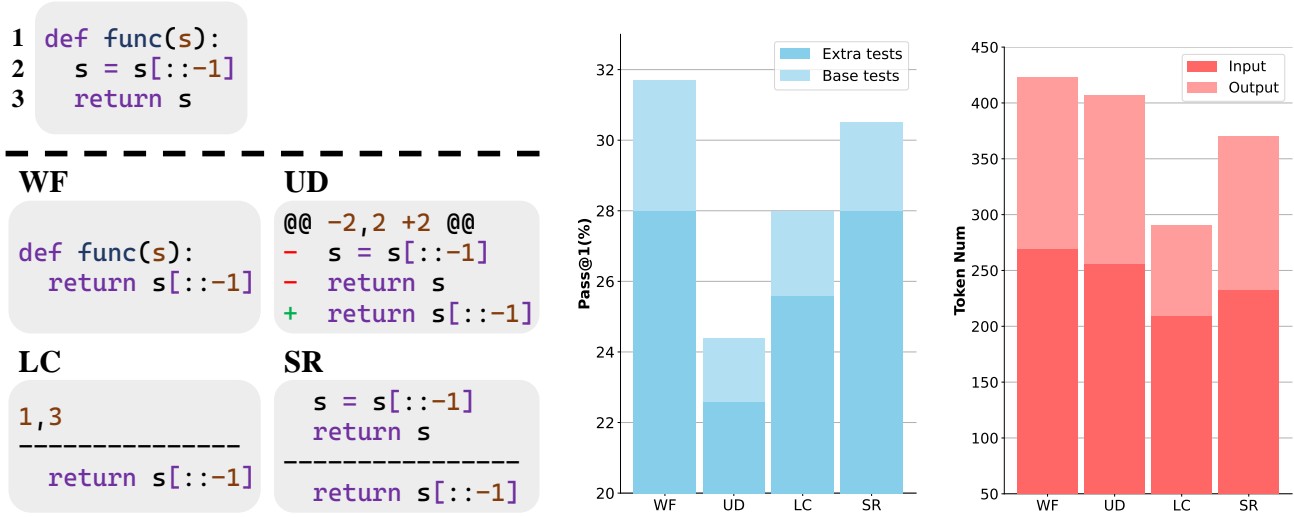

Figure 9. Different formats for representing code modifications.

Figure 10. Performance of models using different formats on APEval.

Figure 11. Context length for models using different formats on APEval.

The results show that using WF yields the best performance, followed by SR and LC, with UD performing the worst. In terms of token usage, LC uses the fewest tokens, followed by SR and UD, while WF uses the most. The average token count for SR and UD is only slightly lower than that of WF, as they are more concise for small code changes, when a large portion needs modification, they must include both versions, making them less efficient than using WF instead.

Recent research has pointed out correlations and scaling laws between model input and output length, as well as performance (OpenAI, 2024; Snell et al., 2024). Our results align with these findings. As the length increases, performance improves consistently across LC, SR, and WF. UD performs poorly in both token usage and performance, likely because it contains redundant information, such as both line numbers and content for the modified sections, where only one would suffice. This redundancy reduces the format's efficiency compared to the other three formats.

## C. Details regarding the collection process of APEval

We inform the annotators about the function's entry point and its purpose, and allow them to send instructions to the AI programming assistant at appropriate moments. We then use screen recording tools to capture the annotators' process of wrtining this function. Afterward, we manually analyze the recordings to construct our benchmark. The historical information, current code, and user instructions are all provided by annotators based on the specified function functionality, to cover various code editing scenarios.

During the process of creating the benchmark, in order to better evaluate the model's ability to utilize historical edits and

integrate this information with user instructions, we collected samples for the $(H, C)$ and $(H, C, U)$ types that required the use of relevant historical information to accurately infer user intent. If a sample contained only a single type of information (such as only $C$ or only $U$), it might be impossible to provide an adequate answer due to a lack of sufficient information.

In our benchmark collection process, we initially annotated one programming process for each task. For some tasks, the annotators consulted the programming assistant; for others, they did not. Similarly, some tasks involved complex editing histories, while others did not. Upon reviewing the data, we found that for certain tasks, it was nearly impossible to collect realistic programming processes containing specific types of information. For example, Some tasks are straightforward and can be completed with just a few lines of code. Programmers who have undergone basic training can write these solutions quickly without needing to consult an assistant or repeatedly revise their code. Conversely, some tasks may involve calling specific libraries or algorithms that most annotators are unfamiliar with, leading them to rely on the programming assistant. It would be unrealistic and counterproductive to instruct annotators to "always consult the AI" or "edit your code repeatedly," as this would deviate from real-world scenarios and undermine our intention to use human-annotated data. Considering these reasons, we did not collect programming traces for the entire test set. While we still hope that the number of samples of four different combinations is at least balanced. At this stage, the number of samples for combinations involving all four data types was relatively similar. So we asked annotators to label additional programming process traces for combinations with fewer samples and collected the corresponding traces. Meanwhile, for combinations with slightly more samples, we discarded some of their traces. Subsequently, we manually translated them into different programming languages. Through this process, we established our final benchmark. Simplified examples of the annotated data is illustrated in Figure 12.

**Example 1**

```
# Current
def has_close_elements(n, t):
  for i in range(len(n - 1)):
    for j in range(i + 1, len(n)):
      if n[i] - n[j] < t or n[j] - n[i] < t:
```

**Example 2**

```
# History 1
def incr_list(l: list):
  return [x++ for x in l]

# Current
def incr_list(l: list):
```

*Figure 12.* Simplified examples of APEval, which covering various code editing scenarios that require integrating multiple types of information to infer user intent. The left example checks if any two numbers in a list are closer than a given threshold. The current logic is flawed and should verify if the absolute difference between two values is less than $t$. The model must detect this issue, fix the error, and generate the remaining code. The right example shows a programmer replacing incorrect code with a corrected version. Without historical edits, the model cannot infer the function's intent. Thus, it must use edit history to make accurate code edits.

## D. Additional details about Programming-Instruct

In our code editing records, we place no limits on the granularity or number of edits. Changes between two code versions may involve anything from a single character to multiple extensive modifications. However, data collected from various sources may be compressed, resulting in incomplete records. This compression can lead to a higher proportion of large-scale edits, particularly in Git Commit data. To address this issue, we propose a decomposition strategy: when there are multiple changes between versions, we break them down into single-step modifications, with the steps ordered randomly. For Git Commit data, we apply this decomposition strategy with a 90% probability, while for AI Programmer and Online Judge Submission data, we apply it with a 50% probability.

We randomly select a time point from the records to represent $C$. In practice, we prefer the model to provide assistance at earlier stages. Thus, we implement a simple rule where the random selection follows an exponential distribution, with the probability of selecting each time point decreasing by 10% with each subsequent step. This biases the model toward choosing earlier time points.

In addition to generating $H$ and $U$, as discussed in Section 4.2, we also simulate the programmer's specification of the target area and model interactions in a chat-style format. The target modification area is created using a random algorithm, as described in Appendix F, while the chat-style interaction is generated using LLMs which is similar to the generation of instructions. Prompts used for it are provided in Appendix O.

# E. Training details

Our models are trained for 2 epochs using the Transformers library (Wolf et al., 2020). We enhance memory efficiency and speed with techniques such as Deepspeed ZeRO3 (Rajbhandari et al., 2019), ZeRO Offload (Ren et al., 2021), FlashAttention2 (Dao, 2024), and triton kernels (Hsu et al., 2024). We calculate the maximum sequence length that can be processed per batch based on the available VRAM. Using the First-Fit Decreasing algorithm (Kundu et al., 2024), we pack training samples to ensure that each batch reaches its maximum sequence length, thereby optimizing training speed. The training process employs the Adafactor optimizer (Shazeer & Stern, 2018) with a learning rate of 5e-5, coupled with a cosine scheduler featuring 15 warm-up steps.

# F. Target area representation

To modify code, programmers often specify the parts requiring changes, typically in one of two ways: either by clicking with the cursor to indicate a general area or by selecting a specific text range with defined start and end points. We model both cases using special tokens: "$<|target|>$" for cursor positions, and "$<|target\_start|>$" and "$<|target\_end|>$" to mark the selected region's boundaries. While collecting training data, we determine modification locations based on the code differences before and after changes. In real-world applications, the decision to provide explicit locations—and their granularity—varies among programmers. To account for this variability, we introduce randomized choices for determining the form and location, integrating this approach into the Programming-Instruct pipeline.

We evaluate CursorCore-DS-1.3B on APEval both with and without location information to assess its impact on performance. The results in Figure 13 show that including location information has minimal effect, likely because most APEval examples are relatively short, enabling LLMs to easily infer modification locations, much like humans do without a cursor. Previous works, such as those on automated program repair (Zhang et al., 2024a), have emphasized the importance of identifying the modification location. We believe this emphasis stems from traditional code completion and insertion paradigms, as well as the natural alignment of specifying modification points with human thought processes. However, with the advancement of LLMs, the benefit of providing location information diminishes when generating code at the function or file level. This may need further exploration in longer contexts, such as repository-level editing tasks.

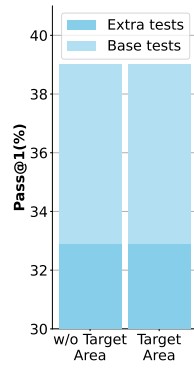

*Figure 13.* With and without the use of location information on APEval.

# G. Discussion about thought process

Incorporating reasoning processes in prompts has been shown to improve model performance, as demonstrated in various works like CoT (Wei et al., 2022) and ReACT (Yao et al., 2023). Some studies have even integrated these processes into the training phase to further enhance effectiveness (Zelikman et al., 2022). In this work, we also explore a self-taught approach, where we prompt LLMs to reverse-generate the reasoning process from outputs and incorporate them into the model's output during training. Our model and data setup follow the same configuration as described in Appendix B to enable quick experiments. The evaluation results are shown in Figure 14.

After incorporating reasoning into training, the model shows slight performance improvements, but the output length increases significantly. The tokens used for reasoning often exceed those in the modified code. Since many programming-assist applications require real-time responses, longer reasoning times may be impractical, so we do not integrate this process into CursorCore. We believe that the decision to use reasoning processes should be based on a combination of factors, such as performance, latency, model size, and specific application requirements.

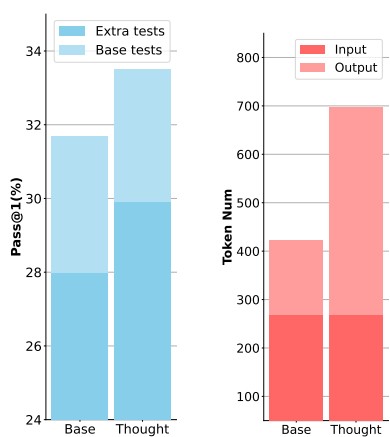

*Figure 14.* Performance of models using thought process or not on APEval.

## H. Data Selection Ablation

We train the smallest model Deepseek-Coder-1.3B on different combinations of datasets to determine the optimal data mix. The results of the ablation study are shown in Figure 15.

**AI Programmer has the highest data quality**   Among the various data sources, the model trained on the AI Programmer dataset achieve the best performance on APEval. We believe this is primarily because the data aligns well with the required format of APEval. Moreover, unlike other data sources such as Git Commit, the AI Programmer data is almost entirely synthesized by LLMs, except for the initial code. As LLMs have advanced, the quality of their generated data has generally surpassed that of data collected and filtered from human-created sources.

**Importance of mixing data with different information types**   We find that using high-quality chat-style data alone, such as the Evol-Instruct dataset, does not achieve the desired performance; it underperforms compared to the AI Programmer dataset. However, when combining both datasets, the model shows a notable improvement. This indicates that to better align the model with a variety of data and information, it is necessary to use datasets containing diverse types of information.

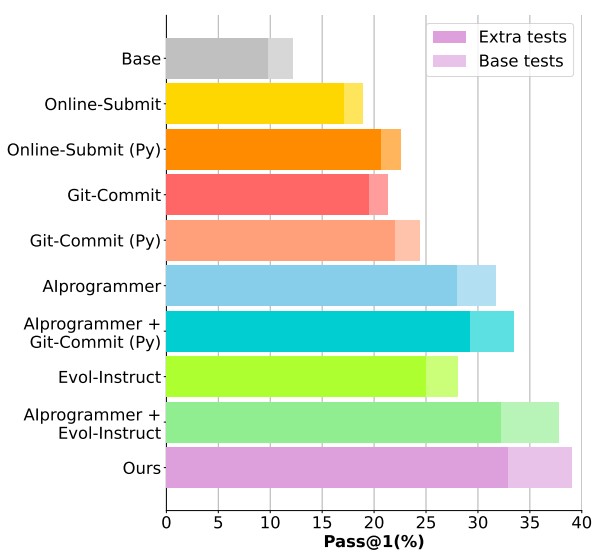

*Figure 15.* Data Selection Ablation on APEval.

**Our final selection**   We combine data from all sources for training. Since current research on Code LLMs primarily focuses on performance in Python, and training on multilingual data leads to a slight decrease in APEval scores, we use only the Python part of the Git Commit and Online Judge Submission datasets. As a result, we get CursorCore series models.

## I. Conversation retrieval for Assistant-Conversation

Not all code editing records are necessary for inferring user intent and predicting output. Some past modifications, such as simple typos corrected shortly after, offer little value to future predictions, and thus can be safely removed. Additionally, if a programmer continuously interacts with the model without deleting these records, the editing history will accumulate and grow until it exceeds the model's maximum context length. This could negatively affect performance and speed.

To address this, it is essential to compress the editing history or retrieve only the relevant portions. Similar to how many conversation retrieval techniques, such as memory modules (Packer et al., 2023), prompt compression (Jiang et al., 2023) and query rewriting (Ye et al., 2023), are used to manage dialogues for chatbots, these methods can be adapted for handling code editing records. In this work, we explore a basic approach, sliding window, to investigate possible solutions. When the number of historical editing records surpasses a predefined threshold, the model automatically discards the oldest entries.

We evaluate this method on APEval, as shown in Figure 16. The impact of setting a sliding window of a certain size on the results is minimal, indicating that compressing the historical records effectively balances performance and efficiency.

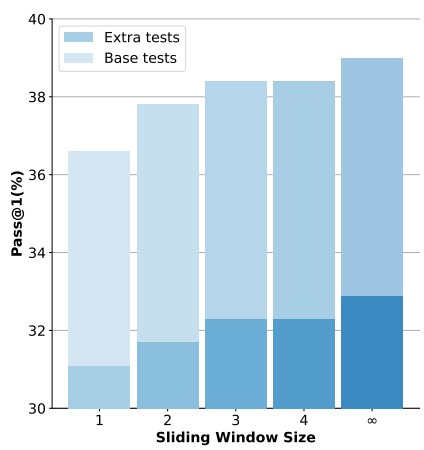

*Figure 16.* Performance of models using different sliding window sizes evaluated on APEval.

*Table 5.* Evaluation results on EvalPlus, CanItEdit and OctoPack.

| Model | EvalPlus | | CanItEdit | | OctoPack |
|---|---|---|---|---|---|
| | HE (+) | MBPP (+) | Desc. | Lazy | HE Fix |
| DS-Coder-6.7B-Base | 47.6 (39.6) | 70.2 (56.6) | 34.3 | 27.6 | 23.8 |
| DS-Coder-6.7B-Inst | 74.4 (71.3) | 75.1 (66.1) | 41.9 | 31.4 | 42.1 |
| CursorCore-DS-6.7B (Chat) | **78.0 (73.2)** | 74.1 (63.8) | **45.7** | 31.4 | **43.3** |
| CursorCore-DS-6.7B (Inline) | 73.8 (67.1) | 71.2 (59.8) | 38.1 | **32.4** | 32.3 |
| CursorCore-DS-6.7B (Tab) | 72.0 (65.9) | 74.3 (63.0) | 6.7 | 6.7 | 25.6 |
| Yi-Coder-9B | 55.5 (47.0) | 69.6 (56.9) | 47.6 | 34.3 | 32.3 |
| Yi-Coder-9B-Chat | 83.5 (76.8) | 84.4 (71.4) | 58.1 | 45.7 | 54.3 |
| CursorCore-Yi-9B (Chat) | **84.1 (79.3)** | **84.4 (73.5)** | 56.2 | 41.0 | **56.1** |
| CursorCore-Yi-9B (Inline) | 79.9 (72.0) | 83.6 (69.6) | 48.6 | 35.2 | 33.5 |
| CursorCore-Yi-9B (Tab) | 79.3 (71.3) | 83.9 (72.5) | 10.5 | 10.5 | 25.6 |
| Qwen2.5-Coder-7B | 61.6 (53.0) | 76.7 (63.0) | 49.5 | 40.0 | 17.1 |
| Qwen2.5-Coder-7B-Inst | 87.2 (83.5) | 83.5 (71.7) | 53.3 | 44.8 | 54.3 |
| CursorCore-QW2.5-7B (Chat) | 80.5 (75.6) | 77.0 (64.3) | 51.4 | **44.8** | 50.6 |
| CursorCore-QW2.5-7B (Inline) | 79.9 (73.2) | 77.0 (64.0) | **57.1** | 39.0 | 41.5 |
| CursorCore-QW2.5-7B (Tab) | 79.9 (74.4) | 75.1 (64.3) | 5.7 | 5.7 | 27.4 |
| DS-Coder-1.3B-Base | 34.8 (26.8) | 55.6 (46.9) | 13.3 | 8.6 | 1.2 |
| DS-Coder-1.3B-Inst | 65.2 (59.8) | 61.6 (52.6) | 26.7 | 17.1 | 29.3 |
| CursorCore-DS-1.3B (Chat) | **68.9 (63.4)** | 61.9 (49.7) | 21.9 | 14.3 | **30.4** |
| CursorCore-DS-1.3B (Inline) | 57.9 (53.7) | 60.1 (51.1) | 25.7 | **17.1** | 17.1 |
| CursorCore-DS-1.3B (Tab) | 63.4 (57.3) | **65.6 (54.8)** | 2.9 | 2.9 | 8.5 |
| Yi-Coder-1.5B | 40.6 (34.8) | 59.0 (50.0) | 21.0 | 12.4 | 3.7 |
| Yi-Coder-1.5B-Chat | 67.7 (64.0) | 66.9 (56.6) | 21.0 | 23.8 | 37.2 |
| CursorCore-Yi-1.5B (Chat) | **68.9 (65.2)** | 65.6 (54.8) | 27.6 | **24.8** | **38.4** |
| CursorCore-Yi-1.5B (Inline) | 60.4 (54.3) | 65.6 (55.0) | **28.6** | **24.8** | 22.6 |
| CursorCore-Yi-1.5B (Tab) | 67.1 (59.1) | 66.1 (**56.6**) | 4.8 | 4.8 | 20.1 |
| Qwen2.5-Coder-1.5B | 43.9 (36.6) | 69.3 (58.5) | 31.4 | 22.9 | 4.9 |
| Qwen2.5-Coder-1.5B-Inst | 70.7 (66.5) | 69.3 (59.4) | 28.6 | 21.0 | 32.9 |
| CursorCore-QW2.5-1.5B (Chat) | **71.3 (65.9)** | 69.3 (58.5) | **31.4** | **22.9** | **36.6** |
| CursorCore-QW2.5-1.5B (Inline) | 66.5 (60.4) | 68.5 (58.2) | 23.8 | 20.0 | **36.6** |
| CursorCore-QW2.5-1.5B (Tab) | 64.0 (58.5) | 67.2 (56.6) | 1.0 | 1.0 | 13.4 |

## J. Evaluation results of other benchmarks

We also evaluate CursorCore on other well-known benchmarks. We use HumanEval+ and MBPP+ (Liu et al., 2023) to evaluate Python program synthesis, CanItEdit (Cassano et al., 2023b) for instructional code editing, and the Python subset of HumanEvalFix from OctoPack (Muennighoff et al., 2024) for automated program repair. All benchmarks are based on their latest versions, and HumanEvalFix uses the test-based repair version as described in the original paper. To generate results, we consistently use vLLM (Kwon et al., 2023) due to its versatility and support for customized conversation formats. Evaluations are conducted within each benchmark's execution environment.

Unlike previous LLMs, CursorCore supports multiple input formats, and different formats may produce different results. To comprehensively showcase this, we categorize input formats based on specific assisted programming scenarios into three cases:

- Chat: Similar to the chat format of ChatGPT (Ouyang et al., 2022), we wrap the query before passing it to the model, which returns a response in a chat style. The final result is obtained after post-processing.

- Inline: Similar to Copilot Inline Chat (Github-Copilot, 2022) and Cursor Command K (Cursor-AI, 2023) scenarios, corresponding to the combination of $C$ and $U$ in Assistant-Conversation. Compared to the Chat mode, it is more tightly integrated with the IDE and returns less additional content.

- Tab: Similar to the use case of Copilot++ (Cursor-AI, 2023), it is the most automated of all scenarios. We provide only the $C$ to the model. For instructional code editing and automated code repair, no explicit instructions are passed.

Evaluation results are shown in Table 5. Our model outperforms the corresponding instruction-tuned and base models across several benchmarks. However, the performance of the 6B+ model, when compared to its corresponding models, is not as strong as that of the 1B+ model. Notably, with the recent release of Qwen2.5-Coder-7B at the start of our experiments, we outperform it on only one benchmark, while other models achieve better performance across more benchmarks. We attribute it to the quantity of high-quality data: larger models require more high-quality data for training. While the current dataset is sufficient to train a highly effective 1B+ model, additional data is needed to train a more competitive 6B+ model.

We analyze the evaluation results of various input types defined in real-world assisted programming scenarios. The results of the Chat and Inline modes are comparable, with Chat mode showing a slight advantage. We attribute this to the flexibility of the Chat format, which allows the model to output its thought process and thus enhances output accuracy. The Tab mode shows comparable results on EvalPlus but underperforms on HumanEvalFix and struggles with CanItEdit, likely due to variations in the informational content of task instructions. For program synthesis based on docstrings, instructions like "complete this function" provide minimal additional context. In contrast, program repair tasks provide crucial information by indicating the presence of errors. When only code is available, the model must first determine correctness independently. Instructional code editing tasks clearly state objectives, such as implementing a new feature, requiring the model to fully understand the given information, as accurate predictions based solely on code are nearly impossible.

To further evaluate the ability of CursorCore to leverage historical information for editing and its applicability to more general software engineering tasks, we additionally conduct experiments on Zeta (Zed-Industries, 2025), DS1000 (Lai et al., 2022), and ClassEval (Du et al., 2023), as shown in Table 6. For Zeta, we report the average accuracy across all evaluated samples, with correctness judged by GPT-4o based on the associated assertion text. For DS1000 and ClassEval, we choose to use the Inline and Tab modes, as they most

*Table 6.* Evaluation results on Zeta, DS1000 and ClassEval.

| Model | Zeta | DS1000 | ClassEval |
|---|---|---|---|
| DS-Coder-1.3B-Base | 18.2 | 16.2 | 13.0 |
| DS-Coder-1.3B-Inst | 42.4 | 20.7 | 13.0 |
| CursorCore-DS-1.3B | **45.5** | **21.2** | **17.0** |

closely resemble the original formats of them. We report the average score across all samples, using the subset of ClassEval that evaluates class-level generation. All generations are produced under greedy decoding. These results collectively demonstrate the strong effectiveness of CursorCore.

## K. Additional evaluation results on APEval

We also report the evaluation results of various versions of other well-known models on APEval, as shown in Table 7.

## L. Multilingual evaluation results on APEval

We report the evaluation results on multilingual versions of APEval, as shown in Tables 8 to 13. CursorCore series achieve state-of-the-art performance across all languages, strongly demonstrating the effectiveness of our approach.

## M. Chat template

Our model's chat template (OpenAI, 2023) is adapted from the ChatML template, where each message in the conversation is restricted to one of the following roles: system, history, current, user, or assistant. The assistant's output includes both code modifications and chat interaction with the user. To indicate code changes, we use two special tokens "<|next_start|>" and "<|next_end|>" to wrap the code modification parts. This approach models Assistant-Conversation effectively and is compatible with standard ChatML templates and chatbot applications. Figure 17 illustrates an example of our chat template, while Figure 18 presents examples of the chat template when using the LC and SR modes described in Appendix B.

## N. Prompts for evaluation

We report the prompts used to evaluate base LLMs on APEval in Table 20, while the prompts used for evaluating instruct LLMs are presented in Table 21.

*Table 7.* Additional evaluation results of LLMs on APEval.

| Model | C | H, C | C, U | H, C, U | Total |
|---|---|---|---|---|---|
| Llama-3.2-1B | 0.0 (0.0) | 14.6 (12.2) | 2.4 (4.9) | 14.6 (12.2) | 7.9 (7.3) |
| Llama-3.2-1B-Instruct | 7.3 (7.3) | 14.6 (14.6) | 19.5 (19.5) | 22.0 (19.5) | 15.9 (15.2) |
| Gemma-2-2B | 7.3 (7.3) | 4.9 (2.4) | 12.2 (12.2) | 14.6 (9.8) | 9.8 (7.9) |
| Llama-3.2-3B | 14.6 (14.6) | 12.2 (9.8) | 26.8 (19.5) | 22.0 (17.1) | 18.9 (15.2) |
| StarCoder2-3B | 19.5 (19.5) | 19.5 (17.1) | 22.0 (19.5) | 22.0 (17.1) | 20.7 (18.3) |
| Phi-3.5-3.8B-Inst | 24.4 (22.0) | 19.5 (14.6) | 34.1 (34.1) | 39.0 (34.1) | 29.3 (26.2) |
| StarCoder2-7B | 7.3 (7.3) | 14.6 (12.2) | 19.5 (14.6) | 22.0 (17.1) | 15.9 (12.8) |
| Llama-3.1-8B | 12.2 (12.2) | 17.1 (14.6) | 19.5 (19.5) | 22.0 (17.1) | 17.7 (15.9) |
| Gemma-2-9B | 22.0 (22.0) | 19.5 (17.1) | 17.1 (19.5) | 22.0 (17.1) | 20.1 (18.9) |
| Codegeex4-All-9B | 43.9 (41.5) | 34.1 (31.7) | 73.2 (61.0) | 34.1 (34.1) | 46.3 (42.1) |
| StarCoder2-15B | 26.8 (24.4) | 24.4 (22.0) | 43.9 (36.6) | 29.3 (24.4) | 31.1 (26.8) |
| DS-Coder-V2-16B-Base | 24.4 (24.4) | 22.0 (19.5) | 31.7 (26.8) | 22.0 (17.1) | 25.0 (22.0) |
| DS-Coder-V2-16B-Inst | 43.9 (41.5) | 41.5 (31.7) | 68.3 (63.4) | 36.6 (31.7) | 47.6 (42.1) |
| Gemma-2-27B | 36.6 (36.6) | 24.4 (22.0) | 56.1 (46.3) | 26.8 (24.4) | 36.0 (32.3) |
| Gemma-2-27B-It | 63.4 (56.1) | 48.8 (41.5) | 68.3 (63.4) | 41.5 (39.0) | 55.5 (50.0) |
| DS-Coder-33B-Base | 31.7 (31.7) | 26.8 (22.0) | 43.9 (36.6) | 24.4 (24.4) | 31.7 (28.7) |
| Llama-3.1-70B | 24.4 (24.4) | 24.4 (22.0) | 46.3 (39.0) | 29.3 (24.4) | 31.1 (27.4) |
| Llama-3.1-70B-Inst | 61.0 (56.1) | 46.3 (46.3) | 65.9 (58.5) | 56.1 (51.2) | 57.3 (53.0) |
| Qwen2.5-72B | 63.4 (61.0) | 36.6 (34.1) | 75.6 (63.4) | 39.0 (34.1) | 53.7 (48.2) |
| DS-Coder-V2-236B-Base | 41.5 (39.0) | 36.6 (31.7) | 58.5 (56.1) | 36.6 (34.1) | 43.3 (40.2) |
| GPT-4o-Mini | 17.1 (17.1) | 36.6 (31.7) | 78.0 (70.7) | 53.7 (43.9) | 46.3 (40.9) |

## O. Prompts for data collection

We design specific system prompts and few-shot examples to collect high-quality training data, as we find that many examples are very difficult to complete with current LLMs, and only a few of them can be successfully completed using rough prompts. For AI Programmer, we utilize LLMs to simulate programmers at three different skill levels, with each level using a distinct set of prompts as shown in Tables 14 to 16. Additionally, prompts used for evaluating whether the outputs align with user intent, generating user instructions, and facilitating chat interactions between models and users are outlined in Tables 17 to 19. Partial few-shot examples are shown in Figures 19 to 24.

## P. Limitations and future work

**Repo-level development assistance**    In this work, we focus on supporting the development of single files or function-level code. However, real-world development operates at the repository level, involving multiple files and greater interaction with IDEs. Previous research has made notable advances in repository-level tasks such as code completion (Zhang et al., 2023), issue fixing (Jimenez et al., 2024), and documentation generation (Luo et al., 2024a). Repository-level code assistance deals with larger datasets, and achieving optimal performance and speed will require more effort. We leave the exploration of multi-file repository-level programming assistance and leveraging additional IDE interactions for future work.

**More scenarios and criteria for evaluation**    Our benchmark is relatively small and based on a multilingual extension of HumanEval, making it insufficient to cover all development scenarios. Beyond using the classic Pass@k metric to evaluate accuracy, other criteria should also be considered, such as evaluating the model's efficiency, security, and redundancy (Huang et al., 2024; Pearce et al., 2021; Li et al., 2024a).

**Preference-based optimization**    Methods like PPO (Schulman et al., 2017) and DPO (Rafailov et al., 2023), which optimize models based on human preferences, have been widely used in LLMs. In programming assistance, programmers can provide feedback on predicted outputs for identical or similar coding processes, further optimizing the model (Shinn et al., 2023). To enable this, a significant amount of feedback data from programmers using AI-assisted tools should be collected or synthesized.

*Table 8.* Evaluation results of LLMs on the C++ version of APEval.

| Model | C | H, C | C, U | H, C, U | Avg. |
|---|---|---|---|---|---|
| **6B+ Models** | | | | | |
| DS-Coder-6.7B-Base | 41.5 | 36.6 | 58.5 | 31.7 | 42.1 |
| DS-Coder-6.7B-Inst | 29.3 | 34.1 | 61.0 | 39.0 | 40.9 |
| Yi-Coder-9B | 29.3 | 43.9 | 34.1 | 29.3 | 34.1 |
| Yi-Coder-9B-Chat | 34.1 | 36.6 | 56.1 | 39.0 | 41.5 |
| Qwen2.5-Coder-7B | 39.0 | 41.5 | 65.9 | 39.0 | 46.3 |
| Qwen2.5-Coder-7B-Inst | 12.2 | 31.7 | 63.4 | 41.5 | 37.2 |
| CursorCore-DS-6.7B | 48.8 | 43.9 | 65.9 | 34.1 | 48.2 |
| CursorCore-Yi-9B | **53.7** | **46.3** | **68.3** | 39.0 | **51.8** |
| CursorCore-QW2.5-7B | 46.3 | 43.9 | 63.4 | **43.9** | 49.4 |
| **1B+ Models** | | | | | |
| DS-Coder-1.3B-Base | 29.3 | 4.9 | 34.1 | 9.8 | 19.5 |
| DS-Coder-1.3B-Inst | 31.7 | 39.0 | 41.5 | 36.6 | 37.2 |
| Yi-Coder-1.5B-Chat | 24.4 | 26.8 | 17.1 | 22.0 | 22.6 |
| Yi-Coder-1.5B | 19.5 | 22.0 | 14.6 | 14.6 | 17.7 |
| Qwen2.5-Coder-1.5B | 34.1 | 31.7 | 48.8 | 36.6 | 37.8 |
| Qwen2.5-Coder-1.5B-Inst | 17.1 | 31.7 | 34.1 | 34.1 | 29.3 |
| CursorCore-DS-1.3B | 39.0 | 39.0 | 56.1 | **39.0** | 42.7 |
| CursorCore-Yi-1.5B | **46.3** | 34.1 | 56.1 | 34.1 | 42.7 |
| CursorCore-QW2.5-1.5B | 41.5 | **39.0** | **58.5** | 36.6 | **43.9** |

*Table 9.* Evaluation results of LLMs on the Java version of APEval.

| Model | C | H, C | C, U | H, C, U | Avg. |
|---|---|---|---|---|---|
| **6B+ Models** | | | | | |
| DS-Coder-6.7B-Base | 51.2 | 43.9 | 61.0 | 46.3 | 50.6 |
| DS-Coder-6.7B-Inst | 48.8 | 34.1 | 68.3 | 51.2 | 50.6 |
| Yi-Coder-9B | 43.9 | 46.3 | 36.6 | 41.5 | 42.1 |
| Yi-Coder-9B-Chat | 43.9 | 41.5 | 56.1 | 39.0 | 45.1 |
| Qwen2.5-Coder-7B | 63.4 | 53.7 | 70.7 | 46.3 | 58.5 |
| Qwen2.5-Coder-7B-Inst | 22.0 | 46.3 | 70.7 | 51.2 | 47.6 |
| CursorCore-DS-6.7B | 63.4 | 56.1 | 65.9 | 48.8 | 58.5 |
| CursorCore-Yi-9B | 56.1 | **61.0** | 68.3 | 46.3 | 57.9 |
| CursorCore-QW2.5-7B | **65.9** | 58.5 | **78.0** | **51.2** | **63.4** |
| **1B+ Models** | | | | | |
| DS-Coder-1.3B-Base | 29.3 | 24.4 | 36.6 | 26.8 | 29.3 |
| DS-Coder-1.3B-Inst | 39.0 | 39.0 | 48.8 | 43.9 | 42.7 |
| Yi-Coder-1.5B | 24.4 | 29.3 | 24.4 | 36.6 | 28.7 |
| Yi-Coder-1.5B-Chat | 31.7 | 24.4 | 12.2 | 26.8 | 23.8 |
| Qwen2.5-Coder-1.5B | 39.0 | 43.9 | 53.7 | 48.8 | 46.3 |
| Qwen2.5-Coder-1.5B-Inst | 34.1 | 36.6 | 53.7 | 48.8 | 43.3 |
| CursorCore-DS-1.3B | 43.9 | 43.9 | **58.5** | 43.9 | 47.6 |
| CursorCore-Yi-1.5B | **53.7** | 48.8 | 53.7 | 46.3 | 50.6 |
| CursorCore-QW2.5-1.5B | **53.7** | **51.2** | 53.7 | **51.2** | **52.4** |

Table 10. Evaluation results of LLMs on the JavaScript version of APEval.

| Model | C | H, C | C, U | H, C, U | Avg. |
|---|---|---|---|---|---|
| **6B+ Models** | | | | | |
| DS-Coder-6.7B-Base | 29.3 | 26.8 | 51.2 | 39.0 | 36.6 |
| DS-Coder-6.7B-Inst | 48.8 | 41.5 | 58.5 | 36.6 | 46.3 |
| Yi-Coder-9B | 22.0 | 41.5 | 17.1 | 39.0 | 29.9 |
| Yi-Coder-9B-Chat | 48.8 | 34.1 | 56.1 | 41.5 | 45.1 |
| Qwen2.5-Coder-7B | 51.2 | 36.6 | 68.3 | 43.9 | 50.0 |
| Qwen2.5-Coder-7B-Inst | 22.0 | 34.8 | 63.4 | 43.9 | 40.9 |
| CursorCore-DS-6.7B | 48.8 | 41.5 | 61.0 | 36.6 | 47.0 |
| CursorCore-Yi-9B | **56.1** | **51.2** | **70.7** | 43.9 | **55.5** |
| CursorCore-QW2.5-7B | 51.2 | 36.6 | **70.7** | **46.3** | 51.2 |
| **1B+ Models** | | | | | |
| DS-Coder-1.3B-Base | 19.5 | 9.8 | 26.8 | 26.8 | 20.7 |
| DS-Coder-1.3B-Inst | 29.3 | 34.1 | 36.6 | 36.6 | 34.1 |
| Yi-Coder-1.5B | 19.5 | 17.1 | 7.3 | 22.0 | 16.5 |
| Yi-Coder-1.5B-Chat | 17.1 | 19.5 | 9.8 | 12.2 | 14.6 |
| Qwen2.5-Coder-1.5B | 31.7 | 31.7 | 48.8 | 39.0 | 37.8 |
| Qwen2.5-Coder-1.5B-Inst | 19.5 | 26.8 | 43.9 | 39.0 | 32.3 |
| CursorCore-DS-1.3B | **34.1** | **36.6** | 41.5 | **43.9** | 39.0 |
| CursorCore-Yi-1.5B | **34.1** | 34.1 | 41.5 | 41.5 | 37.8 |
| CursorCore-QW2.5-1.5B | **34.1** | 34.1 | **51.2** | 41.5 | **40.2** |

Table 11. Evaluation results of LLMs on the Go version of APEval.

| Model | C | H, C | C, U | H, C, U | Avg. |
|---|---|---|---|---|---|
| **6B+ Models** | | | | | |
| DS-Coder-6.7B-Base | 26.8 | 24.4 | 31.7 | 39.0 | 30.5 |
| DS-Coder-6.7B-Inst | 29.3 | 39.0 | 56.1 | 43.9 | 42.1 |
| Yi-Coder-9B | 39.0 | 29.3 | 39.0 | 36.6 | 36.0 |
| Yi-Coder-9B-Chat | 34.1 | 24.4 | 41.5 | 31.7 | 32.9 |
| Qwen2.5-Coder-7B | 41.5 | 36.6 | 53.7 | 41.5 | 43.3 |
| Qwen2.5-Coder-7B-Inst | 12.2 | 22.0 | 58.5 | 46.3 | 34.8 |
| CursorCore-DS-6.7B | 56.1 | 34.1 | 56.1 | 46.3 | 48.2 |
| CursorCore-Yi-9B | 48.8 | 34.1 | 56.1 | 36.6 | 43.9 |
| CursorCore-QW2.5-7B | **58.5** | **41.5** | **58.5** | **51.2** | **52.4** |
| **1B+ Models** | | | | | |
| DS-Coder-1.3B-Base | 22.0 | 7.3 | 26.8 | 29.3 | 21.3 |
| DS-Coder-1.3B-Inst | 31.7 | 22.0 | 36.6 | 34.1 | 31.1 |
| Yi-Coder-1.5B | 7.3 | 14.6 | 2.4 | 9.8 | 8.5 |
| Yi-Coder-1.5B-Chat | 19.5 | 14.6 | 17.1 | 29.3 | 20.1 |
| Qwen2.5-Coder-1.5B | 34.1 | 24.4 | 39.0 | 36.6 | 33.5 |
| Qwen2.5-Coder-1.5B-Inst | 22.0 | 22.0 | 36.6 | 34.1 | 28.7 |
| CursorCore-DS-1.3B | 41.5 | **34.1** | 43.9 | **41.5** | 40.2 |
| CursorCore-Yi-1.5B | 46.3 | 29.3 | 43.9 | 34.1 | 38.4 |
| CursorCore-QW2.5-1.5B | **51.2** | **34.1** | **61.0** | **41.5** | **47.0** |

Table 12. Evaluation results of LLMs on the Rust version of APEval.

| Model | C | H, C | C, U | H, C, U | Avg. |
|---|---|---|---|---|---|
| **6B+ Models** | | | | | |
| DS-Coder-6.7B-Base | 26.8 | 26.8 | 31.7 | 39.0 | 31.1 |
| DS-Coder-6.7B-Inst | 29.3 | 31.7 | 34.1 | 39.0 | 33.5 |
| Yi-Coder-9B | 29.3 | 34.1 | 29.3 | 39.0 | 32.9 |
| Yi-Coder-9B-Chat | 26.8 | 29.3 | 46.3 | 31.7 | 33.5 |
| Qwen2.5-Coder-7B | 46.3 | 34.1 | 48.8 | 36.6 | 41.5 |
| Qwen2.5-Coder-7B-Inst | 9.8 | 22.0 | 51.2 | 36.6 | 29.9 |
| CursorCore-DS-6.7B | 36.6 | 34.1 | 34.1 | 39.0 | 36.0 |
| CursorCore-Yi-9B | 41.5 | 34.1 | 48.8 | **41.5** | 41.5 |
| CursorCore-QW2.5-7B | **48.8** | **39.0** | **51.2** | 39.0 | **44.5** |
| **1B+ Models** | | | | | |
| DS-Coder-1.3B-Base | 22.0 | 22.0 | 31.7 | 29.3 | 26.2 |
| DS-Coder-1.3B-Inst | 19.5 | 26.8 | 36.6 | 31.7 | 28.7 |
| Yi-Coder-1.5B | 22.0 | 24.4 | 17.1 | 29.3 | 23.2 |
| Yi-Coder-1.5B-Chat | 14.6 | 22.0 | 12.2 | 29.3 | 19.5 |
| Qwen2.5-Coder-1.5B | 34.1 | 29.3 | 39.0 | 41.5 | 36.0 |
| Qwen2.5-Coder-1.5B-Inst | 26.8 | 24.4 | 36.6 | 36.6 | 31.1 |
| CursorCore-DS-1.3B | 24.4 | 29.3 | 36.6 | 34.1 | 31.1 |
| CursorCore-Yi-1.5B | 24.4 | **31.7** | **41.5** | 31.7 | 31.1 |
| CursorCore-QW2.5-1.5B | **36.6** | **31.7** | 39.0 | **43.9** | **37.8** |

Table 13. Average evaluation results of LLMs across different language versions on APEval.

| Model | C | H, C | C, U | H, C, U | Avg. |
|---|---|---|---|---|---|
| **6B+ Models** | | | | | |
| DS-Coder-6.7B-Base | 34.2 | 30.9 | 45.9 | 36.2 | 36.8 |
| DS-Coder-6.7B-Inst | 40.3 | 37.0 | 58.1 | 40.6 | 44.0 |
| Yi-Coder-9B | 32.1 | 37.0 | 28.9 | 35.8 | 33.4 |
| Yi-Coder-9B-Chat | 40.6 | 34.2 | 54.9 | 36.6 | 41.5 |
| Qwen2.5-Coder-7B | 49.6 | 40.7 | 62.2 | 39.8 | 48.1 |
| Qwen2.5-Coder-7B-Inst | 16.7 | 33.8 | 63.8 | 43.5 | 39.5 |
| CursorCore-DS-6.7B | 53.7 | 41.9 | 58.5 | 40.2 | 48.6 |
| CursorCore-Yi-9B | 51.7 | **45.5** | **64.6** | 41.9 | 50.9 |
| CursorCore-QW2.5-7B | **56.1** | 43.5 | **64.6** | **46.7** | **52.7** |
| **1B+ Models** | | | | | |
| DS-Coder-1.3B-Base | 20.3 | 13.4 | 28.9 | 23.6 | 21.5 |
| DS-Coder-1.3B-Inst | 31.9 | 33.3 | 39.9 | 36.2 | 35.3 |
| Yi-Coder-1.5B | 16.7 | 19.1 | 13.8 | 22.0 | 17.9 |
| Yi-Coder-1.5B-Chat | 22.4 | 17.9 | 19.5 | 23.2 | 20.7 |
| Qwen2.5-Coder-1.5B | 36.2 | 31.3 | 46.8 | 39.9 | 38.5 |
| Qwen2.5-Coder-1.5B-Inst | 22.4 | 26.4 | 41.5 | 37.4 | 31.9 |
| CursorCore-DS-1.3B | 36.6 | 37.0 | 48.4 | 38.2 | 39.9 |
| CursorCore-Yi-1.5B | 41.8 | 35.3 | 50.8 | 37.4 | 41.1 |
| CursorCore-QW2.5-1.5B | **43.9** | **39.8** | **54.9** | **42.3** | **45.2** |

**Integrated Development Environment (IDE)**

H₁
```python
1  frequency[c] = frequency[c] + 1
```

H₂
```python
1  def function(s):
2    for c in s:
3      frequency[c] = frequency[c] + 1
```

H₃
```python
1  def function(s):
2    for c in s:
3      frequency[c] += 1
```

C
```python
1  def function(s):
```

U
```
Implement it concisely.
```
```python
1  def function(s):
```

A
```python
1  def function(s):
2    frequency = {c: s.count(c) for c in set(s)}
3    return frequency
```
```python
1  def function(s):
2    frequency = {c: s.count(c) for c in set(s)}
3    return frequency
```

**Chat Template**

```
<|im_start|>system
You are a helpful programming assistant.<|im_end|>
<|im_start|>history
```python
frequency[c] = frequency[c] + 1
```<|im_end|>
<|im_start|>history
```python
def function(s):
  for c in s:
    frequency[c] = frequency[c] + 1
```<|im_end|>
<|im_start|>history
```python
def function(s):
  for c in s:
    frequency[c] += 1
```<|im_end|>
<|im_start|>current
```python
def function(s):
```<|im_end|>
<|im_start|>user
```python
Implement it concisely.
```<|im_end|>
<|im_start|>assistant
<|next_start|>```python
def function(s):
  frequency = {c: s.count(c) for c in set(s)}
  return frequency
```<|next_end|><|im_end|>
```

*Figure 17.* Example of chat template and its corresponding demonstration in the IDE scenario.

**Enhance performance with API calls**   We aim to integrate function calls (Patil et al., 2023) into the model to further enhance its capabilities. One potential application is incorporating function calls into the thinking process, such as retrieving information or executing partial code for feedback. Although our final models excludes this thinking step due to performance and speed considerations, we are exploring hybrid approaches to introduce this process while maintaining speed and combine it with other strategies for searching how to edit. Another application is leveraging function calls in output, where calling a Python script for tasks like variable replacement might be more efficient than manually generating code blocks or search-and-replace strategies. For repository-level changes, using terminal commands or IDE APIs could sometimes be a more convenient solution.

**Expand to other applications**   Our framework is designed for programming assistance applications, but the alignment approach can also be applied to other types of AI assistants. For example, in designing an art assistant, it should be able to predict the next drawing step based on the artist's previous drawing patterns, the current state of the canvas, and the artist's instructions. Extending this approach to design assistants for other applications is an interesting research direction.

**Chat Template (LC)**

```
<|im_start|>system
You are a helpful programming assistant.<|im_end|>
<|im_start|>history
0,3
```python
frequency[c] = frequency[c] + 1
```<|im_end|>
<|im_start|>history
2,3
```python
    frequency[c] = frequency[c] + 1
```<|im_end|>
<|im_start|>history
1,1
```python
  for c in s:
    frequency[c] += 1
```<|im_end|>
<|im_start|>current
```python
1 def function(s):
```<|im_end|>
<|im_start|>user
```python
Implement it concisely.
```<|im_end|>
<|im_start|>assistant
<|next_start|>1,1
```python
  frequency = {c: s.count(c) for c in set(s)}
  return frequency
```<|next_end|>
We uses a dict comprehension to count the
occurrences. It is more concise and readable
compared to before.<|im_end|>
```

**Chat Template (SR)**

```
<|im_start|>system
You are a helpful programming assistant.<|im_end|>
<|im_start|>history
```python
frequency[c] = frequency[c] + 1
<|search_and_replace|>
def function(s):
  for c in s:
    frequency[c] = frequency[c] + 1
```<|im_end|>
<|im_start|>history
```python
    frequency[c] = frequency[c] + 1
<|search_and_replace|>
    frequency[c] += 1
```<|im_end|>
<|im_start|>history
```python
def function(s):
  for c in s:
    frequency[c] += 1
<|search_and_replace|>
def function(s):
```<|im_end|>
<|im_start|>current
```python
def function(s):
```<|im_end|>
<|im_start|>user
```python
Implement it concisely.
```<|im_end|>
<|im_start|>assistant
<|next_start|>```python
def function(s):
<|search_and_replace|>
def function(s):
  frequency = {c: s.count(c) for c in set(s)}
  return frequency
```<|next_end|>
We uses a dict comprehension to count the
occurrences. It is more concise and readable
compared to before.<|im_end|>
```

*Figure 18.* Example of chat templates in LC and SR modes.

*Table 14.* Prompt designed to leverage LLMs for simulating the behavior of a novice programmer.

Please play the role of a novice programmer. You are required to write a piece of code. Simulate the real process of repeatedly adding, deleting, and modifying the code. Please return the code block after each step of editing. While writing the code, make some mistakes, such as incorrect logic or syntax errors, etc.

*Table 15.* Prompt designed to leverage LLMs for simulating the behavior of an ordinary programmer.

Please act as an ordinary programmer. Now, you need to write a piece of code. Please simulate the process of repeatedly adding, deleting, and modifying the code during the actual coding process. Please return the code block after each editing step. Try to simulate the coding process of an ordinary programmer as much as possible.

*Table 16.* Prompt designed to leverage LLMs for simulating the behavior of an expert programmer.

Please play the role of an expert programmer. You are now required to write a piece of code. Please simulate the process of repeatedly adding, deleting, and modifying code during the real coding process. Please return the code block after each step of editing. During the coding process, you should be as professional as possible.

*Table 17.* Prompt designed to generate user instructions.

You are a programming assistant. The following content includes information related to your programming assistance, which may contain the record of the programming process, the current code, the git commit after all changes, relevant details about the problem, and your predicted modifications. Please generate an instruction for you to make the corresponding modifications, ensuring it resembles instructions typically given by a human programmer. The instruction may be detailed or concise and may or may not specify the location of the modification. Return the generated instruction in the following format:
```
**instruction:**
{instruction}
```

*Table 18.* Prompt designed to generate chat-style interactions between models and users.

You are a programming assistant. The following content includes information related to your programming assistance, which may contain the record of the programming process, the current code, the user instruction, and your predicted modifications. Please provide the chat conversation for making the prediction. This may include analyzing the past programming process, speculating on the user's intent, and explaining the planning and ideas for modifying the code. Return your chat conversation in the following format:
```
**chat:**
{chat}
```

*Table 19.* Prompt designed to evaluate whether the outputs align with user intent.

You are tasked with assisting a programmer by maintaining a record of the programming process, including potential future changes. Your role is to discern which changes the programmer desires you to propose proactively. These should align with their actual intentions and be helpful. To determine which changes align with a programmer's intentions, consider the following principles:

1. **Understand the Context**: Assess the overall goal of the programming project. Ensure that any proposed change aligns with the project's objectives and the programmer's current focus.

2. **Maintain Clear Communication**: Before proposing changes, ensure that your suggestions are clear and concise. This helps the programmer quickly understand the potential impact of each change.

3. **Prioritize Stability**: Avoid proposing changes that could introduce instability or significant complexity unless there is a clear benefit. Stability is often more valued than optimization in the early stages of development.

4. **Respect the Programmer's Preferences**: Pay attention to the programmer's coding style and preferences. Propose changes that enhance their style rather than contradict it.

5. **Incremental Improvements**: Suggest changes that offer incremental improvements rather than drastic overhauls, unless specifically requested. This approach is less disruptive and easier for the programmer to integrate.

6. **Consider Long-Term Maintenance**: Propose changes that improve code maintainability and readability. This includes refactoring for clarity, reducing redundancy, and enhancing documentation.

7. **Balance Proactivity and Reactivity**: Be proactive in suggesting improvements that are likely to be universally beneficial (e.g., bug fixes, performance enhancements). However, be reactive, not proactive, in areas where the programmer's specific intentions are unclear or where personal preference plays a significant role.

For each potential change, return 'True' if suggesting this change would be beneficial to the programmer, return 'False' if the change does not align with the programmer's intentions or if they do not want you to predict this change. Give your decision after analyzing each change. Provide your response in the following format:

```
**Analysis of change 1:**

Your analysis here.

**Decision:** 'True' or 'False'

**Analysis of change 2:**

Your analysis here.

**Decision:** 'True' or 'False'

...
```

*Table 20.* Prompt used to evaluate base LLMs.

Read the following messages during programming and return the modified code in this format:

<|next_start|>{modified code}<|next_end|>

<|messages_start|>Programming process 1:
```python
a = 1
b = 2
c = a + b
```

Current code:
```python
i = 1
b = 2
c = a + b
```

User instruction:
Please change variable names.<|messages_end|>

<|next_start|>```python
i = 1
j = 2
k = i + j
```<|next_end|>

Read the following messages during programming and return the modified code in this format:

<|next_start|>{modified code}<|next_end|>

<|messages_start|>Programming process 1:
{Programming process 1}

...

Programming process n:
{Programming process n}

Current code:
{Current code}

User instruction:
{User instruction}<|messages_end|>

*Table 21.* Prompt used to evaluate instruct LLMs.

---

**user**
Read the following messages during programming and return the modified code in this format:

<|next_start|>{modified code}<|next_end|>

Programming process 1:
```python
a = 1
b = 2
c = a + b
```

Current code:
```python
i = 1
b = 2
c = a + b
```

User instruction:
Please change variable names.

**assistant**
<|next_start|>```python
i = 1
j = 2
k = i + j
```<|next_end|>

**user**
Read the following messages during programming and return the modified code in this format:

<|next_start|>{modified code}<|next_end|>

Programming process 1:
{Programming process 1}

...

Programming process n:
{Programming process n}

Current code:
{Current code}

User instruction:
{User instruction}

**assistant**

---

**user:**
**The final code:**
```
def if_continuous_modify(code1, code2, code3):
    """
    Check if code3 is a continuous modification of
code1 and code2.

    Args:
        code1 (str): The first code string.
        code2 (str): The second code string.
        code3 (str): The third code string.

    Returns:
        bool: True if code3 is a continuous
modification of code1 and code2, False otherwise.
    """
    # Calculate the Levenshtein distance between
code1 and code2
    dist1 = Levenshtein.distance(code1, code2)
    # Calculate the Levenshtein distance between
code2 and code3
    dist2 = Levenshtein.distance(code2, code3)
    # Calculate the Levenshtein distance between
code1 and code3
    dist3 = Levenshtein.distance(code1, code3)

    # Check if code3 is a continuous modification of
code1 and code2
    if dist3 == dist1 + dist2:
        return True
    else:
        return False
```

**assistant:**
Alright, let's simulate the development process of this code,
including typical errors and revisions. I'll start from the initial
attempt and evolve the code step by step to reach the final version
you provided.

### Step 1: Initial Attempt
The initial attempt might be a rough structure without importing
necessary libraries and possible syntax errors.

```python
def is_continuous_modify(code1, code2, code3):
    """
    Checks if code3 is a continuation of
modifications from code1 through code2
    """
    dist1 = levenstein.distance(code1, code2)
    dist2 = levenstein.distance(code2, code3)
    dist3 = levenstein.distance(code1, code3)

    if dist3 == dist1 + dist2
        return true
    else:
        return False
```
**Issues:**
- Misspelling of the function name (`Levenshtein` as
`levenstein`).
- Syntax error with missing colon at the end of the `if` statement.
- Incorrect capitalization of `true`.

### Step 2: Syntax Corrections
Addressing the syntax errors and some spelling issues.

```python
def is_continuous_modify(code1, code2, code3):
    """
    Checks if code3 is a continuation of
```

modifications from code1 through code2
```
    """
    dist1 = Levenshtein.distance(code1, code2)
    dist2 = Levenshtein.distance(code2, code3)
    dist3 = Levenshtein.distance(code1, code3)

    if dist3 == dist1 + dist2:
        return True
    else:
        return False
```
**Issues:**
- The `Levenshtein` library is not imported.

### Step 3: Importing the Necessary Library
Adding the import statement for the Levenshtein library.

```python
import Levenshtein

def is_continuous_modify(code1, code2, code3):
    """
    Checks if code3 is a continuation of
modifications from code1 through code2
    """
    dist1 = Levenshtein.distance(code1, code2)
    dist2 = Levenshtein.distance(code2, code3)
    dist3 = Levenshtein.distance(code1, code3)

    if dist3 == dist1 + dist2:
        return True
    else:
        return False
```
**Issues:**
- Could improve naming and documentation clarity.

### Step 4: Renaming Function and Improving Documentation
Final adjustments include renaming the function and enhancing the
docstring for better clarity.

```python
import Levenshtein

def if_continuous_modify(code1, code2, code3):
    """
    Check if code3 is a continuous modification of
code1 and code2.

    Args:
        code1 (str): The first code string.
        code2 (str): The second code string.
        code3 (str): The third code string.

    Returns:
        bool: True if code3 is a continuous
modification of code1 and code2, False otherwise.
    """
    dist1 = Levenshtein.distance(code1, code2)
    dist2 = Levenshtein.distance(code2, code3)
    dist3 = Levenshtein.distance(code1, code3)

    if dist3 == dist1 + dist2:
        return True
    else:
        return False
```

This demonstrates a typical progression in coding, from an initial
rough draft with errors, through corrections and improvements, to a
final, refined code block.

*Figure 19.* Few-shot prompts designed to leverage LLMs for simulating the behavior of a novice programmer.

**user:**
**The final code:**
```
def apply_rotary_pos_emb(q, k, cos, sin,
position_ids, unsqueeze_dim=1):
    cos = cos[position_ids].unsqueeze(unsqueeze_dim)
    sin = sin[position_ids].unsqueeze(unsqueeze_dim)

    b, h, s, d = q.shape
    q = q.view(b, h, s, d // 2, 2).transpose(4,
3).reshape(b, h, s, d)

    b, h, s, d = k.shape
    k = k.view(b, h, s, d // 2, 2).transpose(4,
3).reshape(b, h, s, d)

    q_embed = (q * cos) + (rotate_half(q) * sin)
    k_embed = (k * cos) + (rotate_half(k) * sin)
    return q_embed, k_embed
```

**assistant:**
Let's simulate the coding process to reach the final version of the `apply_rotary_pos_emb` function.

### Step 1: Initial Function Setup
Initially, the programmer might start by defining the function and setting up the parameters they know they need to work with.

```python
def apply_rotary_pos_emb(q, k, cos, sin,
position_ids):
    # Initial setup
    pass
```

### Step 2: Add Basic Operations
Next, they'll add the basic operations that are essential for the function, such as selecting the cosine and sine values based on position IDs.

```python
def apply_rotary_pos_emb(q, k, cos, sin,
position_ids):
    cos = cos[position_ids]
    sin = sin[position_ids]
```

### Step 3: Introduce Unsqueeze Operation
Then, realizing they need to match the dimensions, they add the `unsqueeze` operation.

```python
def apply_rotary_pos_emb(q, k, cos, sin,
position_ids):
    cos = cos[position_ids].unsqueeze(1)
    sin = sin[position_ids].unsqueeze(1)
```

### Step 4: Reshape and Process q and k
After ensuring the dimensions of `cos` and `sin`, the programmer starts processing `q` and `k`. They need to reshape these tensors to work with the embeddings correctly.

```python
def apply_rotary_pos_emb(q, k, cos, sin,
position_ids):
```

```
    cos = cos[position_ids].unsqueeze(1)
    sin = sin[position_ids].unsqueeze(1)

    b, h, s, d = q.shape
    q = q.view(b, h, s, d // 2, 2).transpose(4,
3).reshape(b, h, s, d)

    b, h, s, d = k.shape
    k = k.view(b, h, s, d // 2, 2).transpose(4,
3).reshape(b, h, s, d)
```

### Step 5: Define `rotate_half` and Compute Embeddings
Next, they realize they need a helper function `rotate_half` and compute the final embeddings for `q` and `k`.

```python
def rotate_half(x):
    return torch.cat([-x[..., 1::2], x[..., ::2]],
dim=-1)

def apply_rotary_pos_emb(q, k, cos, sin,
position_ids):
    cos = cos[position_ids].unsqueeze(1)
    sin = sin[position_ids].unsqueeze(1)

    b, h, s, d = q.shape
    q = q.view(b, h, s, d // 2, 2).transpose(4,
3).reshape(b, h, s, d)

    b, h, s, d = k.shape
    k = k.view(b, h, s, d // 2, 2).transpose(4,
3).reshape(b, h, s, d)

    q_embed = (q * cos) + (rotate_half(q) * sin)
    k_embed = (k * cos) + (rotate_half(k) * sin)
    return q_embed, k_embed
```

### Step 6: Final Adjustments for Flexibility
Lastly, realizing the need for more flexibility, they introduce the `unsqueeze_dim` parameter.

```python
def apply_rotary_pos_emb(q, k, cos, sin,
position_ids, unsqueeze_dim=1):
    cos = cos[position_ids].unsqueeze(unsqueeze_dim)
    sin = sin[position_ids].unsqueeze(unsqueeze_dim)

    b, h, s, d = q.shape
    q = q.view(b, h, s, d // 2, 2).transpose(4,
3).reshape(b, h, s, d)

    b, h, s, d = k.shape
    k = k.view(b, h, s, d // 2, 2).transpose(4,
3).reshape(b, h, s, d)

    q_embed = (q * cos) + (rotate_half(q) * sin)
    k_embed = (k * cos) + (rotate_half(k) * sin)
    return q_embed, k_embed
```

This completes the journey to the final version of the `apply_rotary_pos_emb` function.

*Figure 20.* Few-shot prompts designed to leverage LLMs for simulating the behavior of an ordinary programmer.

**user:**
**The final code:**
```
static void replace_all(std::string & s, const
std::string & search, const std::string & replace) {
    std::string result;
    for (size_t pos = 0; ; pos += search.length()) {
        auto new_pos = s.find(search, pos);
        if (new_pos == std::string::npos) {
            result += s.substr(pos, s.size() - pos);
            break;
        }
        result += s.substr(pos, new_pos - pos) +
replace;
        pos = new_pos;
    }
    s = std::move(result);
}
```

**assistant:**
Let's simulate the development process of the `replace_all` function in C++ step by step, showing how a programmer might iteratively refine and improve the code.

### Step 1: Basic Function Structure
Initially, I might start with the simplest possible structure to replace occurrences of a substring with another substring in a given string.

```cpp
void replace_all(std::string & s, const std::string
& search, const std::string & replace) {
    // Initial approach to find and replace
    size_t pos = s.find(search);
    while (pos != std::string::npos) {
        s.replace(pos, search.length(), replace);
        pos = s.find(search, pos + replace.length());
    }
}
```

### Step 2: Consider Performance and Correctness
I realize the initial approach can be inefficient due to modifying the string in place. Also, it doesn't handle overlapping substrings well. Let's use a buffer to build the result.

```cpp
void replace_all(std::string & s, const std::string
& search, const std::string & replace) {
    std::string result;
    size_t pos = 0;
    size_t new_pos;
    while ((new_pos = s.find(search, pos)) !=
std::string::npos) {
        result += s.substr(pos, new_pos - pos);
        result += replace;
        pos = new_pos + search.length();
    }
    result += s.substr(pos);
    s = result;
}
```

### Step 3: Optimize with `std::move`
I notice that the final assignment to `s` can benefit from `std::move` to avoid unnecessary copying.

```cpp
void replace_all(std::string & s, const std::string
& search, const std::string & replace) {
    std::string result;
    size_t pos = 0;
    size_t new_pos;
    while ((new_pos = s.find(search, pos)) !=
std::string::npos) {
        result += s.substr(pos, new_pos - pos);
        result += replace;
        pos = new_pos + search.length();
    }
    result += s.substr(pos);
    s = std::move(result);
}
```

### Step 4: Refine and Transition to Static Function
To ensure encapsulation and promote reusability, I decide to make the function `static`.

```cpp
static void replace_all(std::string & s, const
std::string & search, const std::string & replace) {
    std::string result;
    size_t pos = 0;
    size_t new_pos;
    while ((new_pos = s.find(search, pos)) !=
std::string::npos) {
        result += s.substr(pos, new_pos - pos);
        result += replace;
        pos = new_pos + search.length();
    }
    result += s.substr(pos);
    s = std::move(result);
}
```

### Step 5: Final Refinements
Finally, I realize the for loop structure can be more concise and less error-prone, integrating the position updates right into the loop structure.

```cpp
static void replace_all(std::string & s, const
std::string & search, const std::string & replace) {
    std::string result;
    for (size_t pos = 0; ; pos += search.length()) {
        auto new_pos = s.find(search, pos);
        if (new_pos == std::string::npos) {
            result += s.substr(pos, s.size() - pos);
            break;
        }
        result += s.substr(pos, new_pos - pos) +
replace;
        pos = new_pos;
    }
    s = std::move(result);
}
```

This final version improves readability and maintains the optimized performance by constructing the result string in a single pass.

*Figure 21.* Few-shot prompts designed to leverage LLMs for simulating the behavior of an expert programmer.

**user:**
**record:**
Revised code changes:
```diff
@@ -1 +1,5 @@
+/**
+ * magical invsqrt function from Quake III code
+ * see: http://www.codemaestro.com/reviews/9
+ */
+
```

Current code:
```c
1 /**
2  * magical invsqrt function from Quake III code
3  * see: http://www.codemaestro.com/reviews/9
4  */
5
```

**change 1:**
```diff
@@ -3,3 +3,19 @@
  * see: http://www.codemaestro.com/reviews/9
  */

+float InvSqrt(float x)
+{
+       float xhalf = 0.5f*x;
+       int i = *(int*)&x;
+       i = 0x5f3759df - (i>>1);
+       x = *(float*)&i;
+       x = x*(1.5f-xhalf*x*x);
+       return x;
+}
+
+int main(void) {
+       int result = InvSqrt(0.00056);
+       printf("Result: %d (should be 42)", result);
+       return result != 42;
+}
+
```

**assistant:**
**Analysis of change 1:**

The change introduces the `InvSqrt` function implementation and a basic test case within the `main` function. This change is appropriate and beneficial. The primary goal is to implement and test the `InvSqrt` function, a well-known algorithm from the Quake III code. Adding the function and a basic test case aligns directly with this goal. The implementation is clear and concise, following common C coding practices. The function's logic is straightforward and well-documented, making it easy to understand. Overall, the proposed change is a logical next step in developing the `InvSqrt` function and ensuring it works correctly.

**Decision:** `True`

**user:**
**record:**
Revised code changes:
```diff
@@ -1 +1,21 @@

+package com.google.gwtjsonrpc.client;
+
+public class VoidResult_JsonSerializer extends
JsonSerializer<VoidResult> {
+  public static final VoidResult_JsonSerializer
INSTANCE =
+      new VoidResult_JsonSerializer();
+
+  private VoidResult_JsonSerializer() {
+  }
```

```
+
+  @Override
+  public void printJson(final StringBuffer sb, final
VoidResult o) {
+    sb.append("{}");
+  }
+
+  @Override
+  public VoidResult fromJson(final Object o) {
+    return VoidResult.INSTANCE;
+  }
+}
+
```

Current code:
```java
1
2
3 package com.google.gwtjsonrpc.client;
4
5 public class VoidResult_JsonSerializer extends
JsonSerializer<VoidResult> {
6   public static final VoidResult_JsonSerializer
INSTANCE =
7       new VoidResult_JsonSerializer();
8
9   private VoidResult_JsonSerializer() {
10   }
11
12   @Override
13   public void printJson(final StringBuffer sb, final
VoidResult o) {
14     sb.append("{}");
15   }
16
17   @Override
18   public VoidResult fromJson(final Object o) {
19     return VoidResult.INSTANCE;
20   }
21 }
22
```

**change 1:**
```diff
@@ -1,3 +1,16 @@
+// Copyright 2008 Google Inc.
+//
+// Licensed under the Apache License, Version 2.0 (the
"License");
+// you may not use this file except in compliance with
the License.
+// You may obtain a copy of the License at
+//
+// http://www.apache.org/licenses/LICENSE-2.0
+//
+// Unless required by applicable law or agreed to in
writing, software
+// distributed under the License is distributed on an
"AS IS" BASIS,
+// WITHOUT WARRANTIES OR CONDITIONS OF ANY KIND,
either express or implied.
+// See the License for the specific language governing
permissions and
+// limitations under the License.

 package com.google.gwtjsonrpc.client;

```

**assistant:**
**Analysis of change 1:**

Without enough information to confirm the use of the Apache License, predicting this change may not align with the programmer's intentions or the project's actual licensing requirements.

**Decision:** `False`

*Figure 22.* Few-shot prompts designed to evaluate whether the outputs align with user intent.

**user:**
Revised code changes:
```diff
@@ -3,6 +3,10 @@
 def create_cnn_model(in_channels, config):
     layers = []
     conv2d = nn.Conv2d(in_channels, config,
kernel_size=3, padding=1)
-    layers += [conv2d, nn.ReLU(inplace=True)]
+    if batch_norm:
+        layers += [conv2d, nn.BatchNorm2d(config)]
+    else:
+        layers += [conv2d]
+    layers += [nn.ReLU(inplace=True)]
     model = nn.Sequential(*layers)
     return model
```

Revised code changes:
```diff
@@ -1,6 +1,6 @@
 import torch.nn as nn

-def create_cnn_model(in_channels, config):
+def create_cnn_model(in_channels, config,
batch_norm=False):
     layers = []
     conv2d = nn.Conv2d(in_channels, config,
kernel_size=3, padding=1)
     if batch_norm:
```

Current code:
```
1 import torch.nn as nn
2
3 def create_cnn_model(in_channels, config,
batch_norm=False):
4     layers = []
5     conv2d = nn.Conv2d(in_channels, config,
kernel_size=3, padding=1)
6     if batch_norm:
7         layers += [conv2d, nn.BatchNorm2d(config)]
8     else:
9         layers += [conv2d]
10    layers += [nn.ReLU(inplace=True)]
11    model = nn.Sequential(*layers)
12    return model
```

Changes in predictions:
```diff
@@ -1,12 +1,11 @@
 import torch.nn as nn

-def create_cnn_model(in_channels, config,
batch_norm=False):
+def create_cnn_model(in_channels, config,
batch_norm):
     layers = []
     conv2d = nn.Conv2d(in_channels, config,
kernel_size=3, padding=1)
     if batch_norm:
-        layers += [conv2d, nn.BatchNorm2d(config)]
+        layers += [conv2d, nn.BatchNorm2d(config),
nn.ReLU(inplace=True)]
     else:
-        layers += [conv2d]
-    layers += [nn.ReLU(inplace=True)]
+        layers += [conv2d, nn.ReLU(inplace=True)]
     model = nn.Sequential(*layers)
     return model
```

**assistant:**

**instruction:**
Update the `create_cnn_model` function to ensure that the `ReLU` activation function is added immediately after the `BatchNorm` layer if `batch_norm` is enabled. Adjust the function signature to remove the default value for the `batch_norm` parameter. The updated code should handle the addition of the `ReLU` layer conditionally based on the `batch_norm` parameter.

**user:**
Current code:
```ruby
1 # frozen_string_literal: true
2 module
Extensions::DeferredWorkflowStatePersistence::Workflo
w; end
3 module
Extensions::DeferredWorkflowStatePersistence::Workflo
w::Adapter; end
4 module
Extensions::DeferredWorkflowStatePersistence::Workflo
w::Adapter::DeferredActiveRecord
5     extend ActiveSupport::Concern
6     included do
7       include Workflow::Adapter::ActiveRecord
8       include InstanceMethods
9     end
10
11    module InstanceMethods
12      def persist_workflow_state(new_value)
13        write_attribute(self.class.workflow_column,
new_value)
14        true
15      end
16    end
17 end
18
```

Git commit message after all changes:
Include WorkflowActiverecord in the state persistence extension.

Changes in predictions:
```diff
@@ -1,10 +1,12 @@
 # frozen_string_literal: true
+require 'workflow_activerecord'
+
 module
Extensions::DeferredWorkflowStatePersistence::Workflo
w; end
 module
Extensions::DeferredWorkflowStatePersistence::Workflo
w::Adapter; end
 module
Extensions::DeferredWorkflowStatePersistence::Workflo
w::Adapter::DeferredActiveRecord
     extend ActiveSupport::Concern
     included do
-      include Workflow::Adapter::ActiveRecord
+      include
WorkflowActiverecord::Adapter::ActiveRecord
       include InstanceMethods
     end
```

**assistant:**
**instruction:**
At the beginning of the file, add the statement `require 'workflow_activerecord'`; On line 7, change `include Workflow::Adapter::ActiveRecord` to `include WorkflowActiverecord::Adapter::ActiveRecord`; Ensure the final code reflects the necessary changes for including `WorkflowActiverecord` in the state persistence extension.

*Figure 23.* Few-shot prompts designed to generate user instructions

**user:**

Revised code changes:
```diff
@@ -14,3 +14,30 @@
         if (row == n) {
             vector<string> board = generateBoard(queens, n);
             solutions.push_back(board);
+        } else {
+            for (int i = 0; i < n; i++) {
+                if (columns.find(i) != columns.end()) {
+                    continue;
+                }
+                int diagonal1 = row - i;
+                if (diagonals1.find(diagonal1) !=
diagonals1.end()) {
+                    continue;
+                }
+                int diagonal2 = row + i;
+                if (diagonals2.find(diagonal2) !=
diagonals2.end()) {
+                    continue;
+                }
+                queens[row] = i;
+                columns.insert(i);
+                diagonals1.insert(diagonal1);
+                diagonals2.insert(diagonal2);
+                backtrack(solutions, queens, n, row + 1,
columns, diagonals1, diagonals2);
+                queens[row] = -1;
+                columns.erase(i);
+                diagonals1.erase(diagonal1);
+                diagonals2.erase(diagonal2);
+            }
+        }
+    }
+
+    vector<string> generateBoard(vector<int> &queens, int n)
```

Revised code changes:
```diff
@@ -3,41 +3,3 @@
     vector<vector<string>> solveNQueens(int n) {
         auto solutions = vector<vector<string>>();
         auto queens = vector<int>(n, -1);
-        auto columns = unordered_set<int>();
-        auto diagonals1 = unordered_set<int>();
-        auto diagonals2 = unordered_set<int>();
-        backtrack(solutions, queens, n, 0, columns,
diagonals1, diagonals2);
-        return solutions;
-    }
-
-    void backtrack(vector<vector<string>> &solutions,
vector<int> &queens, int n, int row, unordered_set<int>
&columns, unordered_set<int> &diagonals1, unordered_set<int>
&diagonals2) {
-        if (row == n) {
-            vector<string> board = generateBoard(queens, n);
-            solutions.push_back(board);
-        } else {
-            for (int i = 0; i < n; i++) {
-                if (columns.find(i) != columns.end()) {
-                    continue;
-                }
-                int diagonal1 = row - i;
-                if (diagonals1.find(diagonal1) !=
diagonals1.end()) {
-                    continue;
-                }
-                int diagonal2 = row + i;
-                if (diagonals2.find(diagonal2) !=
diagonals2.end()) {
-                    continue;
-                }
-                queens[row] = i;
-                columns.insert(i);
-                diagonals1.insert(diagonal1);
-                diagonals2.insert(diagonal2);
-                backtrack(solutions, queens, n, row + 1,
columns, diagonals1, diagonals2);
-                queens[row] = -1;
-                columns.erase(i);
-                diagonals1.erase(diagonal1);
-                diagonals2.erase(diagonal2);
-            }
-        }
-    }
-
     vector<string> generateBoard(vector<int> &queens, int n)
```

Revised code changes:
```diff
@@ -3,3 +3,17 @@
     vector<vector<string>> solveNQueens(int n) {
         auto solutions = vector<vector<string>>();
         auto queens = vector<int>(n, -1);
+        solve(solutions, queens, n, 0, 0, 0, 0);
```

```
+        return solutions;
+    }
+
+    vector<string> generateBoard(vector<int> &queens, int n)
+    {
+        auto board = vector<string>();
+        for (int i = 0; i < n; i++) {
+            string row = string(n, '.');
+            row[queens[i]] = 'Q';
+            board.push_back(row);
+        }
+        return board;
+    }
+};
```

Current code:
```cpp
1 class Solution {
2 public:
3     vector<vector<string>> solveNQueens(int n) {
4         auto solutions = vector<vector<string>>();
5         auto queens = vector<int>(n, -1);
6         solve(solutions, queens, n, 0, 0, 0, 0);
7         return solutions;
8     }
9
10     vector<string> generateBoard(vector<int> &queens, int
n) {
11         auto board = vector<string>();
12         for (int i = 0; i < n; i++) {
13             string row = string(n, '.');
14             row[queens[i]] = 'Q';
15             board.push_back(row);
16         }
17         return board;
18     }
19 };
```

Predicted modifications:
```diff
@@ -5,6 +5,23 @@
         auto queens = vector<int>(n, -1);
         solve(solutions, queens, n, 0, 0, 0, 0);
         return solutions;
+    }
+
+    void solve(vector<vector<string>> &solutions,
vector<int> &queens, int n, int row, int columns, int
diagonals1, int diagonals2) {
+        if (row == n) {
+            auto board = generateBoard(queens, n);
+            solutions.push_back(board);
+        } else {
+            int availablePositions = ((1 << n) - 1) &
(~(columns | diagonals1 | diagonals2));
+            while (availablePositions != 0) {
+                int position = availablePositions & (-
availablePositions);
+                availablePositions = availablePositions &
(availablePositions - 1);
+                int column = __builtin_ctz(position);
+                queens[row] = column;
+                solve(solutions, queens, n, row + 1, columns
| position, (diagonals1 | position) >> 1, (diagonals2 |
position) << 1);
+                queens[row] = -1;
+            }
+        }
+    }
+
     vector<string> generateBoard(vector<int> &queens, int n)
     {
```

**assistant:**

**chat:**

The revised code transitions from using sets to bitwise operations for solving the N-Queens problem, aiming to optimize the algorithm by reducing dynamic memory allocations and improving speed. Initially, the original code utilized sets (`unordered_set<int>`) to track columns and diagonals. The user likely seeks to enhance efficiency, prompting the replacement of these sets with integer bit masks. To achieve this, the `solve` function is adjusted to use bitwise operations for conflict checks, calculating `availablePositions` using bitwise AND and NOT operations to identify valid queen placements. The `__builtin_ctz` function finds the least significant bit set to 1, corresponding to a valid column, and the recursive `solve` call updates bit masks for columns and diagonals after placing a queen. This maintains the backtracking logic while leveraging bitwise operations for performance. The `generateBoard` function remains unchanged, as it simply formats the result. This approach ensures efficient, streamlined code for the N-Queens problem.

*Figure 24.* Few-shot prompts designed to generate chat-style interactions between models and users.

