# OpenReview forum: "CursorCore: Assist Programming through Aligning Anything"
_ICML.cc/2025/Conference — ICML 2025 poster_

### Official Review · Reviewer_qAD7 · 2025-03-13

**Overall Recommendation:** 4

**Summary:**

The core problem this paper addresses is that existing coding benchmarks are incongruent with human development processes. The paper argues that an effective coding assistant should be able to use various types of information available to humans to make edits, rather than simply respond to constrained prompts.

To this end, the paper proposes a new benchmark, APEval that provides. a testbed for AI assistants that can utilize a richer context stream and propose / predict modifications to the code. The benchmark is created by human annotation, and based on HumanEval. The paper proposes a pipeline called Programming-Instruct that uses an LLM parameterized by a persona to simulate a human coder. SLMs trained with this pipeline outperform other SLMs and approach the performance of frontier LLMs (GPT-4o).

**Claims And Evidence:**

Yes.

**Essential References Not Discussed:**

No.

**Experimental Designs Or Analyses:**

I checked the details of the experiments in Section 6.

**Methods And Evaluation Criteria:**

Yes.

**Other Comments Or Suggestions:**

None.

**Other Strengths And Weaknesses:**

The paper provides a resource of significant utility to the community (a testbed for more naturalistic coding assistants), and the evaluation of the proposed pipeline is reasonable.

**Questions For Authors:**

None.

**Relation To Broader Scientific Literature:**

The primary novelty of the paper is the task formulation and the dataset. To my knowledge, there are no datasets that provide the edit stream amid a more naturalistic evaluation of coding assistants. In comparison, datasets like HumanEval (which APEval is based on) and SWE-Bench focus on a more restricted setting in which a single prepared instruction is provided and the final product is evaluated.

**Theoretical Claims:**

Not applicable.

---

> ### Author Rebuttal · Authors · 2025-03-31
>
> Thanks for your review. We sincerely appreciate your recognition of our work.

---

### Official Review · Reviewer_mFUu · 2025-03-16

**Overall Recommendation:** 4

**Summary:**

Current code LLMs typically use only the code context and, optionally, user instruction as input, without considering the code’s development history. In this paper, the authors propose training a model to integrate various types of the information - particularly editing history - along with the current context and user instruction to predict future code edits. For evaluation, they introduce a new benchmark, APEval, to evaluate code capabilities using different combinations of information. For training data, they generate 219K samples via Programming-Instruct, which uses LLMs to automatically synthesize training data from GitHub commits and online judge platforms. Using the collected data, they fine-tune multiple state-of-the-art LLMs and develop the CursorCore model series, which outperforms models of comparable size.

**Claims And Evidence:**

My main concern is that the paper’s central claim - that integrating editing history into LLM training data helps models to learn more effectively from extensive editing data and ultimately become better programming assistants - was not clearly demonstrated in the evaluation. While Table-4 shows that CursorCore models outperform their unadapted counterparts in H+C and H+C+U settings, this improvement could be attributed to the unadapted models’ unfamiliarity with the format of history (H).

If I understand correctly, a direct comparison between C+U and H+C+U is not possible because they involve different subset of problems. Therefore, to show the benefit of incorporating edit history in both training and testing, it is necessary to compare the CursorCore model’s performance with and without H on the same set of problems. For example, if we construct H for the C+U problems and use the new H+C+U for prediction, would this improve CursorCore’s performance over C+U?

**Essential References Not Discussed:**

The paper discusses essential related works.

**Experimental Designs Or Analyses:**

Overall, the experimental design and analysis are sound and valid.

**Methods And Evaluation Criteria:**

Overall, the proposed methods and evaluation criteria make sense to me, except for the following issues:

**Unclear data processing**

(page 5) The statement, “let the LLMs judge whether each segment of M aligns with the user's purpose through principle-driven approaches”, is ambiguous. How can the LLM determine whether a code change aligns with user intent without first inferring that intent? Does this mean the LLM only filters out simple and obvious cases, such as private information, based on predefined principles? The prompt in Table 18 suggests that the LLM is instructed to first assess user intent. If that is the case, the statement should be revised for clarity.


**Limited scope of benchmark**

The APEval benchmark consists mostly of simple function-level problems, as they are extended from HumanEval and contain on average 139/31/19 lines of H/C/U. Since part of the training data come from GitHub commits, I would expect that the fine-tuned model to perform well beyond function-level code generation. It would be better if the benchmark included more complex tasks, such as those from DS1000 (data science tasks) or ClassEval [a] (class-level program synthesis).

[a] ClassEval: A Manually-Crafted Benchmark for Evaluating LLMs on Class-level Code Generation. (ICSE 2024)

update after rebuttal: The authors have conducted further experiments on Zeta, which is a non-contaminated, more realistic dataset.

**Other Comments Or Suggestions:**

* Please ensure the correct capitalization of tool names:
  * Line 326 and later paragraphs: “stack” should be “Stack”, “oss-instruct” should be “OSS-Instruct”, “editpackft” should be “EditPackFT”, “evol-instruct” should be “Magicoder-Evol-Instruct” (as per the citation), “sglang” should be “SGLang”, and similar adjustments for other tool names.

**Other Strengths And Weaknesses:**

N/A

**Questions For Authors:**

1. Can you provide further empirical evidence demonstrating that adding history indeed improves the model’s ability to assist with programming? (see Claims And Evidence)


2. From Appendix J, it appears the CursorCore models trained from Qwen2.5-7B perform worse than Qwen2.5-Coder-7B-Inst on EvalPlus and HumanEvalFix. This contradicts CursorCore’s strong overall performance on APEval, which also extends HumanEval. This is surprising to me - could the authors provide further discussion? It also makes me wonder if APEval has some inherent bias. For instance, the C-only setting might be problematic, as the current code alone may not sufficiently indicate the user’s next intention.


3. Could you please clarify what is meant by “user instruction”? For example, in the context of HumanEval completion, do you transform the docstring into user instruction, or do you leave the docstring as the code context? Also, on page 2, it mentions “User instructions ... generated as feedback based on interactions with external environments (such as a code interpreter)”. Can you clarify what type of feedback can be considered as user instructions?

**Relation To Broader Scientific Literature:**

Key contribution:

- Integrating code commit and edit history into code LLM training using a carefully-designed format, making a novel contribution. Prior work primarily focuses on using a snapshot of current code and user instruction as input, while the best way to represent and utilize edit history remains understudied.
- A training dataset with edit history input collected from an automated data synthesis pipeline. The large dataset (219K) requires significant computation resources to construct and can benefit future research.
- Extensive evaluation across models and edit representation formats. It conducts a comprehensive evaluation across a wide range of models (including close-source and open-source, various sizes), along with extensive ablation studies to support its technical choices (such as different representations of code changes, integrating reasoning traces, and data selection ablation).

**Theoretical Claims:**

N/A

---

> ### Author Rebuttal · Authors · 2025-03-31
>
> Thanks for your review. Please see our detailed response below:
>
> > Claims and Evidence & Q1
>
> Thanks for the suggestion. We can provide the results of removing H from the samples in APEval that contain H, and compare them with the results obtained without removal. This ensures a fair comparison under the same conditions. The evaluation results are as follows:
>
> |w/o H|w H|
> |-|-|
> |35.4 (29.9)|39.0 (32.9)|
>
> The settings are consistent with those used in Appendix I. Additionally, the results presented in Appendix I further support it, as they provide evaluation results under different window lengths of H.
>
> > “let the LLMs judge whether each segment of M aligns with the user's purpose through principle-driven approaches” is ambiguous
>
> Thanks for the detailed review. We did prompt the LLMs to first analyze the user's intent before making a judgment, as shown in Table 18. We have corrected this in the revised version.
>
> > It would be better if the benchmark included more complex tasks
>
> Including more benchmarks is certainly beneficial! While, the prompts in the DS1000 and ClassEval lack historical context. Recently, a new dataset called Zeta has become available on HuggingFace (released just last month, so it was not possible to include it before the ICML deadline). Zeta is constructed from real engineering use cases, and better reflects the distribution of tasks in real-world scenarios. It include H (though not U). We have now conducted evaluations on this benchmark, and some results are as follows:
>
> |Model|Metric|
> |-|-|
> |DS-Coder-1.3B-Base|18.2|
> |DS-Coder-1.3B-Inst|42.4|
> |CursorCore-DS-1.3B|45.5|
> |Qwen2.5-Coder-7B|51.5|
> |Qwen2.5-Coder-7B-Inst|54.5|
> |CursorCore-QW2.5-7B|60.6|
>
> The reported metric is the average accuracy over all evaluation samples. We use GPT-4o to assess the correctness of the generated results based on the assertion texts.
>
> Of course, we can also evaluate DS1000 and ClassEval, to assess the model's ability to leverage U and C. We choose to evaluate the performance of CursorCore on these benchmarks using the Inline and Tab modes, as they most closely resemble the original formats of them. Some results are as follows:
>
> ||DS1000|ClassEval|
> |-|-|-|
> |DS-Coder-1.3B-Base|16.2|13.0|
> |DS-Coder-1.3B-Inst|20.7|13.0|
> |CursorCore-DS-1.3B|21.2|17.0|
>
> The reported metrics are averaged over all samples, with class-level generation evaluated using ClassEval. All generations are performed under the greedy decoding setting.
>
> Appendix J also includes another open-domain code editing benchmark (CanItEdit). These results collectively demonstrate the strong effectiveness of CursorCore.
>
> > Performance of Qwen2.5 and Concerns about bias in APEval (Q2)
>
> We have included a discussion of this issue in lines 1130 to 1135. Moreover, while instruction-tuned models are effective at aligning with prompts for program synthesis and code repair, they struggle to align with various forms of contextual information, such as historical context, which is commonly encountered in programming workflows. Therefore, the strong performance of Qwen2.5-Coder-7B-Inst on the synthesis and repair tasks, coupled with its  weaker performance on APEval, is expected.
>
> Regarding your concern about bias in APEval, we have already taken this into consideration during the annotation process. As described in Appendix C, we ensure that the current code is sufficient to indicate the user's next intent.
>
> > Explanation of user instruction (Q3)
>
> For "user instruction," we refer to general prompts or feedback related to the code. Alternatively, from another perspective, if we consider the historical context as representing changes in the internal state, then the user instruction can be viewed as an external signal. The term is defined broadly—any type of prompt or feedback that may assist programming can be regarded as a user instruction, such as "write a quicksort algorithm," "translate this code into Java," or feedback messages like "Traceback...". In normal conversation templates of instruct models, this is commonly labeled as "user" or "instruction." To maintain compatibility and in consideration of a previous reviewer's suggestion, we choose to use it.
>
> For HumanEval, we employed different modes, as described in lines 1093 to 1128. In the Tab mode, we leave the docstring as part of the code context; in contrast, we transform the docstring into a user instruction. This setting helps align the evaluation inputs more closely with those encountered in real-world applications.
>
> > Typo
>
> We have fixed them. Thanks for your careful review.
>
> We appreciate your review and look forward to your response!

---

> > ### Comment · Reviewer_mFUu · 2025-04-03
> >
> > I appreciate the authors for conducting additional experiments. The new results address my concerns, and I’ve raised my score.

---

### Official Review · Reviewer_g7Ss · 2025-03-22

**Overall Recommendation:** 2

**Summary:**

The paper introduces a new family of models called CursorCore, which enables handling of historical context while making code generation or assistant response predictions. The authors also propose Programming-Instruct which is a framework designed to collect data to train CursorCore with the historical code edit context. The authors evaluate that models on APEval, which is a modified version of the popular HumanEval benchmark to incorporate historical context, and show improvements over base models.

**Claims And Evidence:**

There are many claims made in Section 2.2 such as:

> Although they can utilize C, they fail to capture H, limiting the modeling of future changes in
C, and are incapable of deleting or editing code. Although user instructions and reflection information can be used through comments and assert statements, this capability is weak and unstable.
>

> Prompt engineering can integrate some of this information into existing models, but the impact is limited.
>

However, these are not adequately supported in the evaluations conducted in Section 6. Specifically, the baseline of prompt-engineering to incorporate historical context is missing from the evaluations. Without this role of the target data curation and model training is unclear.

Further, the work focuses primarily on incorporating historical edit context into an LLM’s input. However, no empirical justification for this provided, and it is only assumed that this needed. In fact, experiments in Section 6 show that (H, C) which includes historical context is consistently poorer in performance compared to (C) which does not. Generally speaking, I agree that historical context would be needed under certain scenarios but it seems that not sufficient work was done to identify these cases; I don’t think that HumanEval-like evals would benefit from historical user context.

Finally, it is known that HumanEval suffers from contamination due to overlap with public sources [1].

[1] Matton, Alexandre, et al. "On leakage of code generation evaluation datasets." *arXiv preprint arXiv:2407.07565* (2024).

**Essential References Not Discussed:**

The inclusion of historical user context will also benefit agents for code [1, 2]. I think that the work will greatly increase in value by also discussing their contributions in this context.

[1] Jimenez, Carlos E., et al. "Swe-bench: Can language models resolve real-world github issues?." *arXiv preprint arXiv:2310.06770* (2023).

[2] Wang, Xingyao, et al. "Openhands: An open platform for ai software developers as generalist agents." *The Thirteenth International Conference on Learning Representations*. 2024.

**Experimental Designs Or Analyses:**

Please see my other comments.

**Methods And Evaluation Criteria:**

1. I think APEval does not thoroughly capture the relevance of historical user edits for code generation tasks. So while the fine-tuned models are better than baseline models, it is not clear if the proposed methods are really relevant for real-world use cases. I have elaborated this concern above.
2. The method used to collect user edits from GitHub and online judges is also not reflective of the kind of data that the authors are seeking. Specifically, the authors are looking to collect incomplete code snippets (as in Figure 1) but the sources for training data collection do not contain incomplete code snippets but rather earlier versions of complete code snippets.

**Other Comments Or Suggestions:**

Line 21 - collect → collects; evaluate → evaluates

**Other Strengths And Weaknesses:**

1. The paper is mainly well-written and easy to follow.
2. The authors should consider moving away from HumanEval-style of evaluations as this does not reflect real-world use cases. In fact, I’d like to know if the code examples in Figure 3 are actually from real-world sources; I’d be surprised if they are.

**Questions For Authors:**

1. Under which scenarios do you think it is absolutely necessary to include historical user context? Have you considered experiments to demonstrate the value of historical context in the input to LLLMs?
2. Can you comment on the distribution gap between real-world scenarios and HumanEval-style of benchmarks? These days most code generation evaluations follow either software-engineering (SWE-Bench), repository-level coding (CrossCodeEval) or very hard programming problems (LiveCodeBench). As such HumanEval is an outdated benchmark, and you should consider moving away from this, especially given the focus on historical user context.
3. Do you think it would be useful to conduct a study on utilizing historical user context for agentic workflows?

**Relation To Broader Scientific Literature:**

The paper attempts to present extend the traditional code completion task [1, 2, 3] by introducing historical user edit context. I think this is an interesting direction, though the treatment of this problem in this work is not very thorough as I have discussed above.

[1] Chen, Mark, et al. "Evaluating large language models trained on code." *arXiv preprint arXiv:2107.03374* (2021).

[2] Athiwaratkun, Ben, et al. "Multi-lingual evaluation of code generation models." *arXiv preprint arXiv:2210.14868* (2022).

[3] Guo, Daya, et al. "DeepSeek-Coder: When the Large Language Model Meets Programming--The Rise of Code Intelligence." *arXiv preprint arXiv:2401.14196* (2024).

**Theoretical Claims:**

N/A

---

> ### Author Rebuttal · Authors · 2025-03-31
>
> Thanks for your review. Please see our response below:
>
> > Baseline of prompt-engineering to incorporate H is missing
>
> The reviewer may have misunderstood our experimental setup. Our prompt engineering baseline does include H, as shown in Tables 19 and 20.
>
> > No empirical justification for incorporating H. Experiments show that (H, C) which includes historical context is poorer in performance compared to (C) which does not & Q1
>
> A direct comparison between the C and H+C subsets is impossible, as they involve different problem sets. Detailed APEval annotations are in Appendix C. The H+C subset is more challenging due to ambiguous or irrelevant history, making it harder for models to leverage and resulting in lower performance.
>
> While, as reviewer mFUu suggested, we can remove H from the H-included subset for direct comparison; please see our response to mFUu.
>
> Historical information is essential in many cases. For example, during variable renaming, a model without access to edit history may mistakenly revert the name. Similarly, if a user writes and later deletes a draft, only the edit history reveals its intention.
>
> > No sufficient work to identify these cases
>
> For the training data, we do not need to explicitly label which cases require historical information. In practice, the collected historical data naturally includes both helpful and unhelpful contexts. We must train LLMs to make predictions based on such inputs containing noise.
>
> For the benchmark construction, we identify cases where historical edits are necessary to infer the intended changes, shown in Appendix C.
>
> > APEval does not thoroughly capture the relevance of historical user edits & It would not benefits from historical context
>
> As noted above, we already considered this during annotation. We suspect the reviewer may have misunderstood our approach: we did not simply add historical edits to the original code. Since HumanEval docstrings include rich context like functionality and test cases, doing so would offer little benefit. Instead, we removed all docstrings and only showed annotators the signature and its purpose, making it significantly harder. In this setting, historical context becomes more helpful.
>
> > Collect user edits from GitHub and OJs is not reflective of data authors are seeking
>
> The reviewer is concerned that the collected historical edits may already be “complete,” with no need for further edits. Our data collection pipeline has taken it into consideration. We adopt the following perspectives and methods:
>
> 1. For data from OJs, we retain only the user’s first correct submission and submissions preceding it.
>
> 2. As shown in Section 4.2, we use LLMs to judge whether code changes align with user intent based on historical and current context. If the current version is complete and changes are version updates (e.g., adding auther information), the LLMs are designed to filter out such updates, while meaningful edits are retained.
>
> > HumanEval-style of evaluations suffers from contamination & does not reflect real-world use cases & Q2
>
> The base models’ technical reports confirm HumanEval contamination was addressed. We also cleaned our training data (Section 5.2). Besides, our benchmark inputs were significantly modified which further reduce contamination risk.
>
> While HumanEval-style benchmarks differ from real-world scenarios in data length (function/file vs. full repositories) and use of third-party libraries/tools, we still chose to extend this benchmark because:
>
> 1. At the time of our experiments, no public benchmarks included all contextual elements like historical edits and user instructions. Thus, we chose to extend existing benchmarks as the most efficient way to assess models on basic tasks in this paradigm; directly creating a more complex benchmark might obscure performance differences.
>
> 2. Repository-level benchmarks (e.g., SWE-Bench) target different goals than ours (see historical context for agent).
>
> Furthermore, a newly released dataset Zeta is constructed based on real-world engineering cases with historical edits; please see our response to mFUu.
>
> Examples in Figure 3 are from a real-world source; we chose the shortest one to aid clarity and fit space constraints.
>
> > Should also include human annotation rubric/results in Section 3.1
>
> Thanks, we agree it is important. Due to space limits, it's in the appendix for now, but we will add the detailed discussion to Section 3.1 in the camera-ready version.
>
> > Historical context for agent & Q3
>
> Including such context could help the agent better understand user intent and adapt to coding style! While, our work focuses on single-call LLM usage with strict latency requirements, unlike agentic workflows, which involve multi-turn interactions and prioritize end-to-end performance. Integrating context into such workflows is a great direction for future work.
>
> > Typo
>
> We have fixed it. Thanks for the careful review.
>
> We appreciate your review and look forward to your response!

---

### Decision · Program_Chairs · 2025-05-01

**Decision:**

Accept (poster)

**Comment:**

This paper presents CursorCore, a new series of models that are fine-tuned on Programming-Instruct data points that include different code context including code editing history, current code context and the user instruction. The paper also proposes a new benchmark APEval for assessing model capability to incorporate different types of contexts. All reviewers found the overall idea of incorporating history and the framework to collect such edit history context interesting and useful. But there were some concerns around the generalizability of the current APEval benchmark to more real-world coding contexts, ablations around necessity of history, limited datasets. The new results in the response on the Zeta dataset as well as the additional results on DS1000 and ClassEval helped with the generalizability of the results beyond just APEval. It would be great to incorporate the additional experiments, rebuttal clarifications and suggestions from the reviews in the final version of the paper.